# I2DFormer: Learning Image to Document Attention for Zero-Shot Image Classification

**Muhammad Ferjad Naeem**[1]     **Yongqin Xian**[1]     **Luc Van Gool**[1]     **Federico Tombari**[2,3]

[1] ETH Zürich     [2] TUM     [3] Google

## Abstract

Despite the tremendous progress in zero-shot learning (ZSL), the majority of existing methods still rely on human-annotated attributes, which are difficult to annotate and scale. An unsupervised alternative is to represent each class using the word embedding associated with its semantic class name. However, word embeddings extracted from pre-trained language models do not necessarily capture visual similarities, resulting in poor zero-shot performance. In this work, we argue that online textual documents, e.g., Wikipedia, contain rich visual descriptions about object classes, therefore can be used as powerful unsupervised side information for ZSL. To this end, we propose I2DFormer, a novel transformer-based ZSL framework that jointly learns to encode images and documents by aligning both modalities in a shared embedding space. In order to distill discriminative visual words from noisy documents, we introduce a new cross-modal attention module that learns fine-grained interactions between image patches and document words. Consequently, our I2DFormer not only learns highly discriminative document embeddings that capture visual similarities but also gains the ability to localize visually relevant words in image regions. Quantitatively, we demonstrate that our I2DFormer significantly outperforms previous unsupervised semantic embeddings under both zero-shot and generalized zero-shot learning settings on three public datasets. Qualitatively, we show that our method leads to highly interpretable results where document words can be grounded in the image regions. Code available at https://github.com/ferjad/I2DFormer.

## 1 Introduction

"What does a tiger look like? It is a fierce animal that looks like a scary, big cat with stripes." Tigers are not native to Japan, yet when the travelers coming from China described them in relation to native animals, it inspired a range of historic paintings depicting tigers in Japan. Humans possess an impressive ability to imagine and identify unseen objects from pure language descriptions. In computer vision, the ability to predict unseen classes is called zero-shot learning, which can be achieved by transferring knowledge from seen classes using auxiliary side information (or semantic embeddings) e.g., attributes [22], word embeddings [17], etc. Although remarkable progress has been made, most of prior works [22, 1, 57, 71, 31, 10] rely on human annotated attributes as the side information. While attributes are appealing, they are often costly to annotate [48, 65, 51] and scale to large datasets. Towards unsupervised semantic embeddings [46, 17, 2], word embeddings can be easily obtained from pre-trained language models [35]. Yet, they often do not reflect fine-grained visual similarities, thus limiting the performance [2].

The goal of this work is to learn visually aligned unsupervised semantic embeddings from online textual documents for zero-shot image classification. With the advent of the Internet, the collective knowledge of humans about the world has been distilled into online encyclopedias like Wikipedia. These encyclopedias present a rich source of fine-grained auxiliary information for a model. While

36th Conference on Neural Information Processing Systems (NeurIPS 2022).

the entries (referred to as documents) may describe an object class with rich visual details, they tend to contain a lot of noise. For example, an entry for 'horse' can define its appearance as well as interesting historic events it participated in. While the former is helpful for a visual model, the latter might introduce noise making it challenging to fully exploit this knowledge.

In this work, we propose a novel model **Image to Document Transformer (I2DFormer)** that learns to align image and document pairs with their global representations as well as with token-wise representations, i.e., image patches and document words. As a result, without any image-level language supervision, our model is able to develop an understanding of different parts of an animal, its habitat, etc, leading to a more discriminative semantic embedding. We summarise our contributions as: (1) We propose a novel transformer based framework for ZSL with noisy documents. (2) Our novel Image to Document Attention (I2D Attention) module learns to identify visually discriminative properties in a document leading to a more discriminative semantic embedding. (3) Our model I2DFormer consistently improves the SOTA in unsupervised semantic embeddings on three challenging datasets, i.e., AWA2, CUB and FLO. Moreover, we qualitatively demonstrate that our model learns highly interpretable results. (4) We show that the learned document embedding can be used with any existing ZSL model to significantly improve its performance. To the best of our knowledge, I2DFormer is the first method to learn an attention-based embedding from noisy documents for ZSL without relying on any pretrained part localization model or attribute vocabulary.

## 2    Related Works

**Zero-shot Learning** aims to generalize a model trained on seen classes onto a disjoint set of unseen classes using shared auxiliary information available for both sets [22]. Several methods in this direction learn a compatibility function between the image and the class embedding space [42, 30, 8, 27, 1, 68, 56, 26]. Another competing line of work uses generative models like GANs to learn the feature space of seen and unseen classes [58, 59, 70, 69, 21, 44]. A complementary line of work focuses on learning improved visual-semantic embeddings [24, 68, 18, 7] and training better image encoders [66, 71, 61]. Semantic embeddings are a crucial building block for all of these methods. However, despite its importance, it is a less studied topic. Human labeled attributes [57, 34, 51, 16, 29] have become the de-facto semantic embedding for most methods. However, they are hard and expensive to scale as they require human experts [48, 65, 51].

**Learning semantic embeddings with minimal supervision** aims to use cheap to obtain side information to learn a semantic embedding with minimal label information. Several works have explored using text corpora as an alternative source of semantic embeddings. Some approaches include using word embeddings from pretrained language models [63, 35, 47, 28] and knowledge graphs [52, 19, 5, 30, 27] to encode semantic similarities. Another line of work aims to directly learn semantic embeddings from documents containing information about classes. Earlier works in this direction used TF-IDF [43] to directly embed the document in a joint image space [14]. Successive works have focused on reducing the noise in the document by using predefined attribute vocabulary [3], learning better weights for TF-IDF embeddings [37] or complementing these embeddings with a part detection network [15, 69]. Recent works have incorporated Transformer based language models to directly embed a document to a semantic embedding [20, 6]. However, all these works either learn the semantic embedding against the global image representation or use a pretrained part detector for the human-labeled attributes to filter the relevant details. VGSE [62] instead proposes to directly learn semantic embeddings from images of seen classes and extrapolate them to the unseen classes by measuring their class name similarities. Our model, I2DFormer instead uses both the knowledge in text documents and the images of seen classes to learn a semantic embedding and ZSL model.

**Learning cross-modal attention between image and text** to ground text in images without region level supervision has been a long-studied problem in visual question answering, image captioning, etc. [11, 12, 40, 41]. Methods in this line of work learn a mapping between the region level features from an image and its caption. More recently, Transformers [50] have made a breakthrough in this field with models like ViLBERT [25] and FILIP [41] that learn a cross-modal attention to learn cross modal embeddings. They show that the grounding of text in the image naturally emerges as a by-product [60]. However, these works rely on having access to image-level text which is expensive to obtain. Our model instead addresses the much more challenging problem of learning a cross-modal embedding and attention from images and their class-level text document.

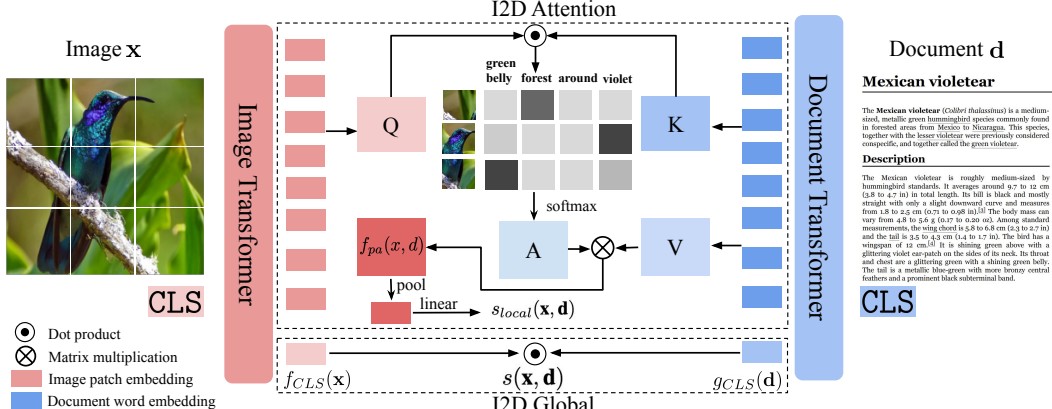

Figure 1: **I2DFormer**, our novel Transformer based model, uses noisy documents as auxiliary information to learn a zero-shot model. The first part of the model, I2D Global, learns to encode images and noisy documents to a shared embedding space. In order to distill discriminative local information from the document, we propose a novel I2D Attention module that learns fine-grained interactions between image patches and document words. Together, the two modules learn a highly discriminative document semantic embedding I2DEmb.

## 3 Image to Document Transformer (I2DFormer)

The vast majority of existing ZSL works utilize either human-annotated attributes or word embeddings as auxiliary information. In this work, we instead utilize the textual collection of encyclopedia (wiki) entries of classes as side information given the wealth of free document collections describing object classes available on the internet. To achieve that, we propose I2DFormer (Figure 1), a pure-transformer ZSL framework that learns to align image and document pairs with their global representations and with token-wise representations i.e., image patches and document words.

**Notations.** We define the classes that are included in the training set as seen classes $\mathcal{Y}^s$, and the classes that are excluded from training as unseen classes $\mathcal{Y}^u$. Let $\mathcal{T} = \{(\mathbf{x}, \mathbf{y}, \mathbf{d}) | \mathbf{x} \in \mathcal{X}^s, \mathbf{y} \in \mathcal{Y}^s, \mathbf{d} \in \mathcal{D}^s\}$ be our training set where $\mathbf{x}$ denotes an RGB image from the training images $\mathcal{X}^s$, $\mathbf{y}$ is its label belonging to the seen classes $\mathcal{Y}^s$, $\mathbf{d}$ is a document e.g., Wikipedia article, containing textual descriptions of the object class $\mathbf{y}$, and $\mathcal{D}^s$ is a collection of documents describing seen classes. At test time, another collection of documents $\mathcal{D}^u$ describing the unseen classes $\mathcal{Y}^u$ will be made available to the model. This simulates an internet query to fetch extra information about an unseen class. Those documents will be used as the side information to connect seen and unseen classes. The task of ZSL is to make a prediction among only unseen classes, while GZSL needs to predict both seen and unseen classes.

### 3.1 I2D Global: Learning joint Image-Document Embeddings with Transformer

Our model is a dual-stream transformer architecture that learns, respectively, an embedding function $\mathcal{F}$, an image transformer [13], for images, and $\mathcal{G}$, a document transformer [50], for text documents. The first part of our model learns a global compatibility between the Image and the Document by our Image to Document(I2D) Global module. On the image side, given an input image $\mathbf{x} \in \mathbb{R}^{H \times W \times C}$, we reshape it into a sequence of flattened 2D patches $\mathbf{x}_p \in \mathbb{R}^{N \times (P^2 \times C)}$, where $(H, W)$ is the size of an input image with $C$ as the RGB channels, $(P, P)$ is the size of each image patch, and $N = HW/P^2$ is the resultant number of patches. Moreover, we append a CLS token to $\mathbf{x}_p$ as the input to the image transformer to learn a global image representation. Inspired by LiT [67], we use a pretrained frozen image transformer [13]. This is followed by a learnable feature projection layer that maps the image embeddings to a joint image-document embedding space with dimensionality $r$. The image encoder $\mathcal{F}$ outputs $f_{CLS}(\mathbf{x}) \in \mathbb{R}^r$ as the global image feature and $f_p(\mathbf{x}) \in \mathbb{R}^{N \times r}$ as the patch-wise image embedding for the input image where $r$ is the feature dimension.

On the document side, given a document $\mathbf{d}$ consisting of $M$ words, we get its token-wise input feature representation with a pretrained word embedding model. Note that we use words and tokens interchangeably as we use GloVe word features as tokens [35]. Since each document consists of

a typically long sequence of words, we further pass this feature representation through a learnable MLP as a token projection layer to reduce the feature dimension and the memory footprint, yielding $\mathbf{d}_t \in \mathbb{R}^{M \times r}$, where $r$ is the feature dimension as the output of the token projection layer. Our learnable document transformer consists of transformer encoder blocks with multi-head attention. We append a CLS $\in \mathbb{R}^r$ token to this sequence and pass it through the document transformer to get $g_{CLS}(\mathbf{d}) \in \mathbb{R}^r$ as the global document embedding and $g_t(\mathbf{d}) \in \mathbb{R}^{M \times r}$ as the word-wise text embedding for the input document. We later refer to the learned $g_{CLS}(\mathbf{d})$ as a document embedding (semantic embedding) I2DEmb that can be used by any ZSL method.

We define a scoring function $s : \mathcal{X} \times \mathcal{D} \to \mathbb{R}$ that measures the similarity of any image $\mathbf{x}$ and document $\mathbf{d}$ pair. The scoring function computes the dot product between global image embedding $f_{CLS}(\mathbf{x})$ and document embedding $g_{CLS}(\mathbf{d})$ , formulated as

$$s(\mathbf{x}, \mathbf{d}) = f_{CLS}(\mathbf{x}) \cdot g_{CLS}(\mathbf{d}). \tag{1}$$

The learning objective is to make the scoring function assign high scores to correct image and document pairs and low scores to incorrect ones. Therefore, for a particular training instance $(\mathbf{x}, \mathbf{y}, \mathbf{d})$, and $\mathcal{D}^s$ the collection of documents belonging to seen classes, we minimize the following cross-entropy loss,

$$L_{CLS} = -\log\left(\frac{\exp s(\mathbf{x}, \mathbf{d})}{\sum_{\mathbf{d}' \in \mathcal{D}^s} \exp s(\mathbf{x}, \mathbf{d}')}\right) \tag{2}$$

### 3.2   I2D Attention: Learning Image Patch to Document Word Attention

Our I2D Global module essentially aligns image-document pairs using their global representations. While this paradigm has been popularized by influential works like CLIP [38], it relies on a large amount of image-text pairs to learn all discriminative local features and represent them in the output of CLS token. However, we are dealing with a more challenging problem where the number of training images is small (a few thousand) and there is only one document associated with each class. Aligning two modalities at a global level will be prone to overfitting and hard to generalize to unseen classes at test time. Moreover, our documents are directly collected from the Internet and therefore are noisy e.g., a large portion of the words are irrelevant to visual appearance. To address these challenges, we propose I2D Attention, a novel cross-modality attention module, to learn fine-grained interaction between image patches and document words, capturing local features defined in the document such as body parts of an animal, their habitat in the form of image background, etc. We argue that learning these local mappings allows a model to generalize beyond the seen classes.

Our I2D Attention module takes as inputs the patch-wise embeddings $f_p(\mathbf{x}) \in \mathbb{R}^{N \times r}$ of the image and the token-wise embeddings $g_t(\mathbf{d}) \in \mathbb{R}^{M \times r}$ of the document. We task the model with searching for the visually-relevant words in the documents using image patches as the queries. More specifically, we define $Q = f_p(\mathbf{x})W_q$ as the image queries, $K = g_t(\mathbf{d})W_k$ as the text keys to compare with, and $V = g_t(\mathbf{d})W_v$ as the text values to mix with after the search, where $W_q$, $W_k$ and $W_v$ are learnable linear transformations, all in size $r \times r$. The I2D Attention module estimates the cross-modal attention $A(\mathbf{x}, \mathbf{d}) \in \mathbb{R}^{N \times M}$ by computing a dot product between every image patch and word pair followed by a softmax,

$$A(\mathbf{x}, \mathbf{d}) = softmax\left(\frac{QK^T}{\sqrt{r}}\right) \tag{3}$$

This attention matrix is used to compute new feature representations $f_{pa}(\mathbf{x}, \mathbf{d}) \in \mathbb{R}^{N \times r}$ for all image patches as linear combinations of rows of the value matrix $V$ i.e., $f_{pa}(\mathbf{x}, \mathbf{d}) = A(\mathbf{x}, \mathbf{d})V$. Intuitively, this operation recomputes the image patch embeddings using the token-wise embeddings of relevant words in a document. To obtain the image-level embedding, we apply global pooling on the patch dimension $N$ of the new patch embeddings $f_{pa}(\mathbf{x}, \mathbf{d})$, yielding $\hat{f}_{pa}(\mathbf{x}, \mathbf{d}) \in \mathbb{R}^{1 \times r}$. Afterwards, we compute the local alignment score between an image-document pair by applying a simple linear layer,

$$s_{local}(\mathbf{x}, \mathbf{d}) = H(\hat{f}_{pa}) \tag{4}$$

where $H \in \mathbb{R}^{r \times 1}$ is a learnable linear layer. Given a particular training example $(\mathbf{x}, \mathbf{y}, \mathbf{d})$, we optimize the following cross-entropy loss,

$$L_{local} = -log\left(\frac{\exp s_{local}(\mathbf{x}, \mathbf{d})}{\sum_{\mathbf{d}' \in \mathcal{D}^s} \exp s_{local}(\mathbf{x}, \mathbf{d}')}\right) \tag{5}$$

We do not use any skip connection similar to previous cross-modal attention blocks like ViLBERT [25] as we want the attention weighted embedding to directly give us a linearly separable representation. Our I2D Attention Module searches for relevant patch features in the document for each training class $y \in \mathcal{Y}_s$ and learns to associate visual concepts with the noisy text in the document. The classification loss $L_{local}$ maximizes the contribution of discriminative words in the document and minimizes the contribution of irrelevant details about a class. Furthermore, calculating cross-entropy over the full seen set ensures that the model is aware of similar attributes between fine-grained classes and can pick additional cues to separate such classes. I2D Attention introduces minimal learnable parameters with only 4 additional linear layers and rather forces $\mathcal{F}$ and $\mathcal{G}$ to minimize irrelevant details in the tokenwise $f_p(\mathbf{x})$ and $g_t(\mathbf{d})$ as well as the global $f_{CLS}(\mathbf{x})$ and $g_{CLS}(\mathbf{d})$ embeddings. This is in contrast to architectures like ViLBERT [25] where several self-attention layers are stacked on top of the cross-modal module to further learn the output embedding with self-attention. We later show in our experiments that this can hurt the performance in our data constrained zero-shot learning setup. Although documents have been explored before in ZSL, prior work uses fixed document embeddings that are encoded with TF-IDF [14, 69, 15] or extracted from a pretrained language model [6, 20]. In contrast, our document embeddings are aided by the attention module to identify important details and thus are also assisted by this additional visual information.

### 3.3 Inference

Given an input image $\mathbf{x}$, we search for the document $\hat{\mathbf{d}}$ that yields the highest compatibility score,

$$\hat{\mathbf{d}} = \underset{\mathbf{d}' \in \mathcal{D}}{\arg\max}\, s(\mathbf{x}, \mathbf{d}'). \tag{6}$$

The search space includes only documents of unseen classes in zero-shot learning i.e., $\mathcal{D} = \mathcal{D}^u$, and all classes in generalized zero-shot learning (GZSL) i.e., $\mathcal{D} = \mathcal{D}^s \cup \mathcal{D}^u$. The final prediction is simply the class label associated with the document $\hat{\mathbf{d}}$. For GZSL, we apply calibrated stacking [9] to calibrate the activations of unseen classes on a held-out set to reduce the bias towards seen classes. We only use the output of the global prediction as it is computationally cheaper and has distilled the knowledge of patch-to-token interactions while training. The attention between image patch and document words is computed as the explainability of the model's decision when required.

## 4  Experiments

We conduct extensive experiments on Animals with Attributes2 (AWA2) [57], Caltech-UCSD Birds (CUB) [51] and Oxford Flowers (FLO) [32], which are widely used datasets in ZSL. We follow the evaluation protocol and data splits proposed by Xian et al. [57]. Since the main focus of this work is to learn unsupervised semantic embeddings, we do not use any human-annotated attributes. In the following, we first describe how documents are collected and implementation details. Then, we quantitatively compare against SOTA unsupervised semantic embeddings methods and ZSL methods. Finally, we show quantitative results to demonstrate the interpretability of our method.

**Collecting documents.** We use online sources for documents that can be queried with minimal human supervision. These sources contain useful knowledge about each class but might have a lot of noise as irrelevant textual details. For AWA2, we use A-Z Animals [53], an animal encyclopedia. For CUB, we use AllAboutBirds [54], a bird-watching encyclopedia. For FLO, we use a collection of gardening blogs and Wikipedia [55] to collect documents for these classes. However, we found documents for flowers to be less focused on the patterns of petals and pistils and rather more focused on the general description of the plant and its taxonomic biological classification. FLO is therefore a challenging dataset to generalize from document-based embeddings. We adopt a simple filtering step on these collected articles similar to [20]. We look at the documents for 10% of classes of each dataset and identify sections that contain relevant information about the class. The rest of the documents are filtered to only contain these sections. The average size of a document is ≈400 words. To put this into perspective, models like CLIP [38, 64] use image captions of at max 64 tokens [38, 36]. The long length of the documents presents an additional challenge. Document examples and their links are included in the supplementary and are available on the github repository.

**Training Details.** We implement our model in PyTorch and train on an Nvidia A100 GPU. We use the VIT/B16 checkpoint trained on ImageNet 1k by [13] as the pretrained Image Transformer. The

| Semantic Embedding | Source | Zero-Shot Learning | | | Generalized Zero-Shot Learning | | | | | | | | |
|---|---|---|---|---|---|---|---|---|---|---|---|---|---|
| | | AWA2 | CUB | FLO | AWA2 | | | CUB | | | FLO | | |
| | | T1 | T1 | T1 | u | s | H | u | s | H | u | s | H |
| GloVe[35] | CLSN | 52.1 | 20.4 | 21.6 | 42.1 | 75.3 | 54.0 | 16.2 | 43.6 | 23.6 | 14.4 | 88.3 | 24.8 |
| GloVe[35] | DOC | 61.6 | 29.0 | 25.8 | 49.5 | 78.1 | 60.6 | 23.8 | **62.6** | 34.5 | 14.7 | 91.0 | 25.3 |
| LongFormer[4] | DOC | 44.2 | 22.6 | 8.8 | 41.6 | 81.8 | 55.2 | 19.9 | 41.0 | 26.8 | 8.8 | 89.8 | 16.0 |
| MPNet[49] | DOC | 61.8 | 25.8 | 26.3 | 58.0 | 76.4 | 66.0 | 20.6 | 44.3 | 28.2 | 22.2 | **96.7** | 36.1 |
| TF-IDF[43] | DOC | 46.4 | 39.9 | 34.0 | 29.6 | **87.6** | 44.2 | 29.0 | 52.1 | 37.3 | 28.9 | 94.8 | 44.3 |
| VGSE[62] | IMG + CLSN | 69.6 | 37.1 | - | 56.9 | 82.8 | 67.4 | 27.6 | 70.6 | 39.7 | - | - | - |
| **I2DFormer**(Ours) | IMG + DOC | **76.4** | **45.4** | **40.0** | **66.8** | 76.8 | **71.5** | **35.3** | 57.6 | **43.8** | **35.8** | 91.9 | **51.5** |

Table 1: **Comparing our I2DFormer with unsupervised semantic embedding methods** using the same image feature and method (our I2D Global module). In ZSL, we report top-1 accuracy (**T1**) on unseen classes, in GZSL on seen/unseen (**s/u**) classes and their harmonic mean (**H**). We consider semantic embeddings that are either directly extracted (with a pretrained language model) or learned from different sources including classnames (CLSN), document (DOC), a combination of image and classnames (IMG+CLSN), and a combination of image and document (IMG+DOC). Our I2DFormer significantly improves on the baselines to set a new SOTA for unsupervised class embeddings.

| Type | ZSL Model | Semantic Embeddings | Zero-Shot Learning | | | Generalized Zero-Shot Learning | | | | | | | | |
|---|---|---|---|---|---|---|---|---|---|---|---|---|---|---|
| | | | AWA2 | CUB | FLO | AWA2 | | | CUB | | | FLO | | |
| | | | T1 | T1 | T1 | u | s | H | u | s | H | u | s | H |
| Discriminative | SJE[2] | GloVe | 56.6 | 27.1 | 13.1 | 41.3 | 83.4 | 55.3 | 14.4 | 51.6 | 22.5 | 4.6 | 93.2 | 8.7 |
| | | VGSE | 70.1 | 31.6 | - | 49.9 | 84.8 | 62.8 | 23.1 | 57.5 | 33.0 | - | - | - |
| | | I2DEmb(Ours) | 72.6 | 38.2 | 33.4 | 55.8 | 82.6 | 66.6 | 25.0 | 56.2 | 34.6 | 18.5 | 87.1 | 30.5 |
| | APN[61] | GloVe | 73.8 | 20.7 | 15.2 | 57.6 | 84.6 | 68.5 | 19.6 | 32.6 | 24.5 | 12.8 | 39.4 | 19.3 |
| | | VGSE | 74.0 | 34.3 | - | 65.0 | 72.4 | 68.5 | 23.2 | 52.9 | 32.1 | - | - | - |
| | | I2DEmb(Ours) | 74.5 | 40.6 | 35.4 | 65.5 | 76.9 | 70.7 | 30.0 | 49.9 | 37.5 | 32.0 | 85.3 | 46.5 |
| Generative | GAZSL[69] | GloVe | 63.7 | 37.5 | 20.9 | 22.2 | 90.8 | 35.6 | 5.93 | 36.2 | 10.2 | 8.38 | 97.3 | 15.4 |
| | | VGSE | 74.7 | 35.7 | - | 29.5 | 93.8 | 44.9 | 10.5 | 51.8 | 10.5 | - | - | - |
| | | I2DEmb(Ours) | 83.1 | 42.9 | 34.2 | 56.8 | **94.7** | 71.0 | 15.9 | 50.4 | 24.1 | 28.8 | 90.1 | 43.7 |
| | f-VAEGAN-D2[59] | GloVe | 70.7 | 31.8 | 32.1 | 65.7 | 69.5 | 67.6 | 23.9 | 55.7 | 33.5 | 25.0 | **99.0** | 39.9 |
| | | VGSE | 75.0 | 40.7 | - | 70.8 | 79.0 | 74.7 | 32.7 | 57.5 | 41.7 | - | - | - |
| | | I2DEmb(Ours) | **85.1** | 41.9 | 36.9 | **73.2** | 81.7 | **77.2** | 33.4 | 57.3 | 42.2 | 30.0 | 97.3 | 45.8 |
| Disc. | **I2DFormer**(Ours) | I2DEmb (Ours) | 76.4 | **45.4** | **40.0** | 66.8 | 76.8 | 71.5 | **35.3** | 57.6 | 43.8 | 35.8 | 91.9 | **51.5** |

Table 2: **Comparing** I2DFormer **with baseline ZSL methods**, under various unsupervised semantic embeddings we see that our model and embeddings I2DEmb set a new SOTA. In ZSL, we report top-1 accuracy (**T1**) on unseen classes, in GZSL on seen/unseen (**s/u**) classes and their harmonic mean (**H**). Best embedding results within a method are underlined. Best results overall are **bolded**.

image patch projection and token projection layers are implemented as a shallow MLP. Maxpool or Meanpool are chosen as global pooling by ablation. The model is trained with Adam optimizer with a learning rate of $1e^{-3}$ and takes $\approx$24 hours to converge. $L_{CLS}$ and $L_{local}$ relative weights are chosen by ablation. More details are available in the supplementary. For baseline methods, we use the CLS features from the same VIT/B16 checkpoint with author's implementations. We ablate these methods over multiple hyperparameters to report the best run. For VGSE, we use the semantic embeddings released by the original authors (not available for FLO).

## 4.1 Comparison with SOTA unsupervised semantic embeddings

In this section, we compare with existing unsupervised semantic embeddings where they are obtained without using human supervision using the same ZSL method (our I2D global module).

**Compared semantic embeddings.** For GloVe (classname) [35], we simply extract GloVe vectors of class names. This method has been adopted by many prior ZSL methods [33, 45, 17, 2, 30] due to its simplicity. For GloVe (Document) [35], we average over the feature vectors of each word in the document. LongFormer [4] is a text transformer model trained for documents and outputs a CLS embedding given a document. MPNet[49] is the current SOTA Sentence Transformer model[39] trained to optimize embeddings for natural language classification tasks. Since the original model is

trained for short sequences, we average over the individual sentence embeddings similar to [20, 6]. `TF-IDF` [43] stands for Term Frequency-Inverse Document Frequency, which has been used by some prior ZSL methods [14, 23]. `VGSE` [62] learns the semantic embeddings from image patches and word embeddings of class names. Since these embedding models generate one embedding for the whole document, we replace the Document Transformer with an equally deep MLP.

**Results.** From Table 1, we observe that our method I2DFormer consistently outperforms all semantic embedding methods in both ZSL and GZSL. Compared to `GloVe` (Document) [35], which also serves as an input to our method (without the average over words), the learned embedding of our model achieves an impressive 76.5% accuracy vs 61.6% on AWA2, 45.4 % vs 29.0 % on CUB and 40.0% vs 25.8% on FLO with a relative $1.25\times$, $1.5\times$ and $1.6\times$ improvement each. This shows that our learned document embedding assisted by our I2D attention module significantly improves the zero-shot performance. We see that these improvements are also consistent in GZSL where we see a significant improvement in the HM. Similar results are observed for other pretrained language semantic embeddings `Longformer`, `MPNet` and `TF-IDF` [4, 49, 43]. Since the original embedding models for these baselines were only trained on language data, the generated semantic embedding is unlikely to capture the most visually discriminative features described in the document. Our model however is able to learn a more informed semantic embedding thanks to supervision from the images of the seen classes. Comparing rows 1 and 2, we see that the use of documents over classnames leads to a major improvement as documents capture better class similarities. Finally, compared to VGSE [62], the current SOTA unsupervised semantic embedding, we observe that our model again substantially outperforms it. While both VGSE and our model exploit patch-wise similarities in images of different classes to learn a class embedding, our model is additionally able to complement this embedding with localized information available from the documents thanks to our I2D Attention.

## 4.2 Comparing with SOTA ZSL methods

In this section, we compare our full model with existing SOTA zero-shot models across baseline embeddings and our learned document embedding. For a fair comparison, we evaluate those methods with the same VIT/B16 image features. We show in Table 2 that our method I2DFormer, and our learned document embeddings `I2DEmb` achieve SOTA performance.

**Compared to baselines**, our model I2DFormer or our learned embedding `I2DEmb` consistently outperform all baseline ZSL methods and embeddings to establish a new SOTA. I2DFormer achieves SOTA ZSL performance on CUB and FLO, the fine-grained datasets. On CUB, I2DFormer achieve an impressive 45.4% compared to the closest 42.9% of GAZSL that also uses our `I2DEmb`. On FLO, I2DFormer achieves 40.0% compared to the closest 36.9% of f-VAEGAN-D2 that again uses our `I2DEmb`. In GZSL, on CUB, I2DFormer achieves 43.8% HM compared to the closest 42.2% of f-VAEGAN-D2 (`I2DEmb`). On FLO, I2DFormer achieves an impressive 51.5% HM compared to the closest 45.8% of f-VAEGAN-D2 (`I2DEmb`). We would like to emphasize that our model is outperforming both the generative baselines in GZSL on these two datasets. Generative models have previously been shown to be the most competitive baselines in these datasets. However, since I2DFormer learn a fine-grained attention between the image patches and the words in the article, it is able to outperform these baselines with this extra knowledge without requiring feature generation. On AWA2, a coarse classification dataset, we see that I2DFormer achieves SOTA performance among the Discriminative baselines. However, the best performance is achieved by the Generative baseline f-VAEGAN-D2 using our `I2DEmb` on this dataset. f-VAEGAN-D2 with `I2DEmb` achieves the best ZSL accuracy of a remarkable 85.1% vs. the closest 83.1% achieved by GAZSL(`I2DEmb`), with I2DFormer being the third. In GZSL, f-VAEGAN-D2 with `I2DEmb` achieves SOTA with an impressive HM of 77.2% followed by 74.7% of the same method with VGSE embeddings. I2DFormer is a second in this setting surpassing the remaining 3 ZSL methods across all embeddings. However, these baselines are only able to outperform I2DFormer with our learned `I2DEmb`.

## 4.3 Ablation study

**What kind of Patch to Word Attention is required in ZSL?** We study the importance of learning patch to word attention for Document based embeddings in Table 3a. We see that while only training I2DGlobal can learn a competitive ZSL model, it significantly improves and achieves SOTA performance with the introduction of our novel I2D Attention module in I2DFormer. We see a

| Model | AWA2 | CUB | FLO |
|---|---|---|---|
| I2DGlobal | 69.4 | 37.2 | 37.2 |
| I2DGlobal + FILIP[64] | 67.3 | 35.7 | 38.3 |
| ViLBERT[25] | 75.0 | 29.9 | 21.3 |
| I2DFormer | **76.4** | **45.4** | **40.0** |

(a)

| Input Embedding | AWA2 | CUB | FLO |
|---|---|---|---|
| LongFormer[4] | 51.5 | 34.3 | 21.8 |
| MPNet[49] | 65.5 | 36.1 | 28.3 |
| Word2Vec[28] | 74.0 | 43.8 | 37.8 |
| GloVe[35] | **76.4** | **45.4** | **40.0** |

(b)

Table 3: **a) Ablation over I2DFormer.** The proposed I2DGlobal module greatly benefits from the addition of I2D Attention to achieve SOTA performance. Comparing against FILIP and VilBERT cross-modal attention, we see that I2D Attention achieves SOTA. **b) Ablating over input embeddings for our Document Transformer** we see that older models like Word2Vec and GloVe serve as better input representation than modern Transformer-based language models.

relative 14%, 16%, and 8% improvement over I2DGlobal. This validates our hypothesis that the patch to word attention distills its knowledge to the global `I2DEmb`, improving its performance. In the same table, we also ablate over 2 competing cross-modal attention modules. FILIP [64] is a recent method that proposes to associate each image patch to its most attended word. We see that this hurts the performance when using noisy Documents. ViLBERT [25] proposes a cross-modal attention module which is paired with a self-attention block [50] to learn an image embedding. We see that while this improves the performance over I2DGlobal on AWA2, it leads to worse performance on our fine-grained datasets CUB and FLO potentially due to the bigger model requiring more training data. Our I2D Attention outperforms both these baselines and achieves SOTA performance.

**What kind of input text representation works best for I2DFormer?** We ablate over several pretrained word/ token representations to be used as an input to our Document Transformer in Table 3b and note that GloVe [35] achieves the best result. We observe that the two Transformer based language models LongFormer [4] and MPNet [49] perform much worse than older baselines Word2Vec [28] and GloVe [35]. We conjecture that this is due to the Document Transformer having limited text data for the seen classes while training. Transformer-based models generate different word features for the same word with self-attention [50]. Documents of unseen classes use the same and additional vocabulary in new sentences causing a distribution shift in their input representation.

## 4.4 Qualitative Results

**Document Transformer attention for `I2DEmb`.** We look at the learned attention over documents of unseen classes in Figure 2a and plot the top 8 most attended words across the Document Transformer attention heads for `I2DEmb`. On AWA2, we see that class name is complemented with human-like labelled attributes for these classes such as the color of the animal, type of the feet, and habitat etc. For the fine-grained datasets, CUB and FLO, we see that for similar classes like the two warblers, the model learns similar attributes like "ruby-crowned" as well as discriminating "tiger stripes" vs "chestnut patterns". We confirm our hypothesis that a learned document embedding will focus on discriminating properties of the class from the noisy document.

**Visualizing Image to Document attention** as the row of the attention matrix in Figure 2b we see that the top 3 words contributing to the $f_p$ for these patch are visually grounded in the image. I2D Attention is able to develop this localization in the image for unseen classes from the noisy documents without any patch, word associated ground truth. We further see that the class name can repeatedly be the most important cue for the model in multiple sentences of the document for some patches.

**Visualizing Document word to Image attention** as the column of the attention matrix in Figure 3, we see the impressive localization ability of I2DFormer for the top attended words in `I2DEmb`. We see that the model is able to localize the unseen classes horse and giraffe in the image despite never observing them while training. The discriminating properties like the hoofed legs are also localized in the image. For CUB, we see that between the two very similar images of two unseen classes, the model identifies the yellow bottom as an important property from the two different documents of the ground truth class. However, the model is further able to identify the discriminative tiger stripes of the Cape May Warbler to differentiate it from the Tropical Kingbird which has gray-green feathers leading to correct classification. Finally, on FLO, the localization ability of our model remains consistent where the Peruvian lily is identified by localizing it as a Lily and identifying its

| | Classname | Top attended words for I2DEmb |
|---|---|---|
| **AWA2** | **Blue Whale** | polar, whale, enormous, water, greyish, massive, blue, migrating |
| | **Sheep** | grass, sleeker, wooly, stocky, rams, horns, sheep, hooves |
| | **Seal** | frigid, fur, hind, cetaceans, pinnipeds, seal, ocean |
| | **Giraffe** | markings, hoofed, giraffe, enormous, mammals, reddish, woodlands, leaves |
| **CUB** | **Green Violetear** | bronzy, bolivia, nicaragua, green chest, canopy, colibri, shining, glittering |
| | **Tropical Kingbird** | gray-headed, kingbird, green, venezuela, whitish, flycatcher, gray-green feathers, plumage |
| | **Cape May Warbler** | yellowish, breast, short-trailed, green, tiger stripes, olive, ruby-crowned, cape |
| | **Chestnutsided Warbler** | chestnut, markings, wingbars, crown, green, ruby-crowned, warbler, oak |
| **FLO** | **Pink Primrose** | buttercup, ranunculus, stigmas, wildflower, primrose, four-petaled, evening blooms, amapola, |
| | **Globe Thistle** | weed, daisy, wooly, thistles, florets, sharp toothed, wrinkled, asteraceae |
| | **Peruvian Lily** | lily, tuber, stripped, curving petals, flecked, alstroemeria, streaked, resupinate |
| | **Tiger lily** | lily, tiger, bulblets, capsules, lilium, tigrinum, pollinated, england |

(a)

(b)

Figure 2: **a) Top attended words for I2DEmb for unseen classes** in the Document Transformer consist of discriminative properties available in the document. **b) Visualizing Image patch to Word attention**, we see that the top 3 most important words are visually grounded in the image.

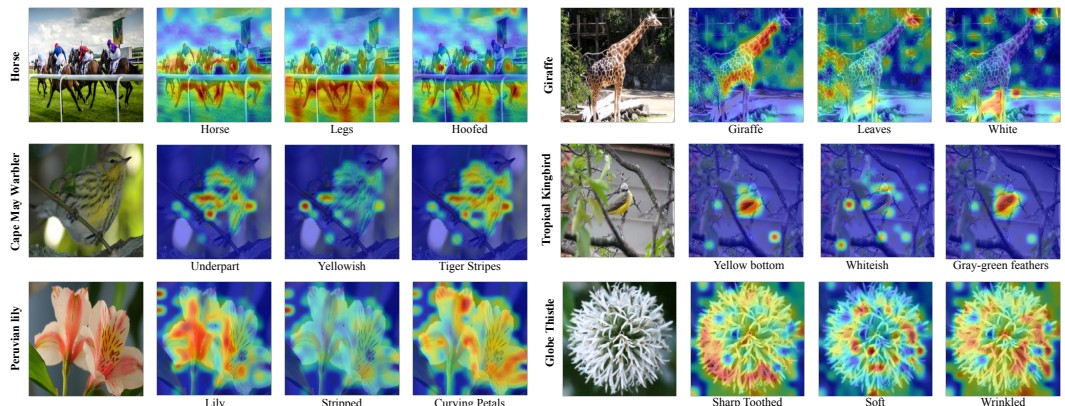

Figure 3: **Visualizing Word to Image Attention** in the most attended document words for I2DEmb, we see that our model I2DFormer has learned to localize them in the image without any paired patch-word supervision. This learned attention differentiates the two similar birds in the second row by identifying and localizing tiger stripes and gray-green as discriminative properties.

stripped and curved petals. Similarly for Globe Thistle, the model is able to differentiate the sharp teeth, soft and wrinkled parts of the flower. The prevalence of these words as top attended words in the document transformer and their impressive localization verifies our hypothesis that the attention module distills its knowledge to the CLS head. A model that does not learn patch to word attention can miss these properties if they are not deemed important among the seen classes.

## 5    Conclusion

We propose I2DFormer, a fully Transformer based framework for learning semantic embeddings from noisy documents. Our I2D Global module learns a shared embedding space between an image and document embeddings. This is assisted by our I2D Attention module learns local features about the class defined in the document without any paired image-level captions. As a result, our full model I2DFormer achieves SOTA performance on both ZSL and GZSL with respect to baseline semantic embedding baselines and zero-shot models. In addition, our model develops an impressive ability to identify and localize discriminative properties of a class from the document in the image. Finally, we show that the learned embeddings from our model can further improve all zero-shot methods. **Broader Impact.** We hope to inspire ZSL works in new fields like medicine, climate change etc. that were limited by expert attribute annotations. Online documents paired with our method can extend the current datasets for these fields. A potential down side of using unsupervised online content is that it can cause unintended biases against gender, race etc. if represented in the collected data. Future works in this direction can study using content moderation tool-kits to clean this knowledge. **Acknowledgements.** Ferjad is supported by a Google Ph.D. Fellowship. We want to thank Wenjia Xu for helpful discussion and support for VGSE related experiments.

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
