# Supplementary Material
# I2DFormer: Learning Image to Document Attention for Zero-Shot Image Classification

**Muhammad Ferjad Naeem**[1]  **Yongqin Xian**[1]  **Luc Van Gool**[1]  **Federico Tombari**[2,3]

[1] ETH Zürich  [2] TUM  [3]Google

In this supplementary, we additionally report the following:

- Section 1.1: Additional ablation results over the two modules of our model.

- Section 1.2: The impact of document sources.

- Section 1.3: Additional qualitative results.

- Section 1.4: The choice of cross-modal attention.

- Section 2: Examples of collected documents.

- Section 3: Additional Training Details.

## 1 Additional Experiments

### 1.1 Ablation between Global and Local Scores.

I2DFormer learns a global score $s$ in the I2DGlobal module and a local score $s_{local}$ in the I2DAttention module. Our comparisons in the main paper use the global score for inference as it is computationally fast and can additionally provide semantic embeddings for other ZSL methods. In the supplementary, we additionally report the performance and ablation with $s_{local}$ in Table 1.

Comparing row 1 and row 2, we see that only training the individual block already results in a competitive model. However, I2D Global achieves better performance as learning cross-modal attention is a harder task than matching global embeddings. Comparing row 2 and 3, we see that combining both modules lead to a major improvement in $s_{local}$ as it distills the knowledge of the global embedding. We see that the two modules of I2DFormer have a symbiotic relationship where both greatly benefit from joint training and achieve a boost in performance. Comparing rows 3 and 4, we see that the global score $s$ achieves better performance as it additionally uses global information of the image and the document and sets the state-of-the-art.

### 1.2 Impact of Document Sources.

In addition to Documents from AllABoutBirds[3], we collect a second set of Documents for CUB from Wikipedia[4] to study the impact of document sources on the performance of I2DFormer. Documents from Wikipedia tend to be on average 50% longer in length, and are more noisy than AllAboutBirds. We observe that I2DFormer achieves a ZSL accuracy of 43.1% with Wikipedia compared to the best 45.4% achieved with AllAboutBirds on CUB. I2DFormer is still able to maintain a competitive performance despite having significantly higher noise in the document, showing the robustness of our method. Note that the performance with Wikipedia documents still outperforms all baselines reported in the main paper Table 1 which use the relatively cleaner AllAboutBirds documents.

36th Conference on Neural Information Processing Systems (NeurIPS 2022).

## 1.3 Additional Qualitative Results.

**Document Transformer attention for** `I2DEmb`**.** In Table 3, we show the top-8 attended words across the Document Transformer attention heads for `I2DEmb` of some unseen classes. We see that the trend from the main paper still holds where each row contains rich discriminative information. We confirm that our method is able to learn document embeddings that focus on discriminative visual words of different classes from their noisy documents.

**Visualizing Document word to Image attention** in Figure 1, we see the impressive localization ability of I2DFormer for the top attended words in `I2DEmb`. The trend from the main paper is consistent across all three datasets. For example, the attention for the word "Horse" (first row) contains the animal and its hoofed legs. The attention for Dolphin (second row), an aquatic mammal, is also well grounded across the word "dolphin", its surrounding "ocean", and the discriminative attribute, "fin". A similar trend is observed in the other two datasets.

**Visualizing Image to Document attention** in Figure 2, we observe that the top 3 words contributing to the $f_{pa}$ for these patches are visually grounded in the image, where $f_{pa}$ is the attention weighted image patch embedding from the document. We see that the impressive localization ability of our model is consistent across all three datasets as it identifies different environments e.g. Pasture and Saltwater; different body parts e.g. head, eye and tail; and patterns of the flowers e.g. Color, Tubular and wide. This further validates the ability of our I2DAttention module to learn visually relevant properties from the noisy document.

**Limitation of learned attention.** I2DFormer utilizes our novel I2DAttention module to develop an impressive localization ability between image and document without any paired supervision for image regions and document words. However, since the model learns this directly from data, it comes with limitations like dataset bias for concepts less represented in the training set. In Figure 3 we show some examples where the model's localization is less optimal. Most images in AWA2 contain only a single instance of the animal. In row 1, we visualize an example for the unseen class Seal with three instances in a single image. We observe that the model misses one of the seals for the word Fur. In row 2, we visualize an instance of woodpecker, an unseen bird class in CUB, which likes to stand vertically to the tree branch. This is in contrast to most instances of birds in the training set where the bird sits parallel to the branch. We observe that the model's attention is spread over larger regions as the model is less confident. An extreme case of this is observed for the word red where the model is focusing on the background more than the bird. Finally in the last row, we see that the model has learned Pollinators to be a strong attribute for this class. However the localization of this attribute by the model is focusing more on the petals compared to the pollinators. However, since our model additionally also learns to match the global features, the impact of localization error can still be minimized by other information available in the document. This also relates to the small difference in accuracy between the $s_{local}$ and $s$ in Table 1.

## 1.4 Direction of Cross-Modal Attention.

Our proposed model I2DFormer utilized our Image to Document(I2D) Attention module. The direction of this attention is motivated by the information asymmetry in our zero-shot problem setting. We have one document for each class while in the image domain, a class can have several thousands training samples. Additionally, a document describes the most discriminative information in the image along with non-visual information about the class. Our I2DAttention learns to focus on the visual information to align the two modalities while learning to limit the impact of non-visual information. An image, however, only contains limited information about the document and does not contain features about the non-visual content of the document. As a consequence, learning Document to Image attention can lead to picking up spurious correlations which can limit the model's performance. We test this emperically in Table 2. Zero-shot Transformer works like CLIP [2] as less likely to be impacted by the same limitation as they train on paired image and captions where a caption locally describes the image.

**Experimental setup.** Baseline a) is our I2DGlobal module introduced in Section 3.1. Baseline (b,c) introduces the Document to Image(D2I) Attention module which is the counter to our I2DAttention module. Baseline b) combines I2DGlobal with D2I Attention similar to our proposed model to learn asymmetric attention from Documents to Image. Baseline c) combines I2DFormer(the proposed

|  |  | Zero-Shot Learning | | | Generalized Zero-Shot Learning | | | | | | | | |
| --- | --- | --- | --- | --- | --- | --- | --- | --- | --- | --- | --- | --- | --- |
|  |  | AWA2 | CUB | FLO | AWA2 | | | CUB | | | FLO | | |
| Model | Scoring | T1 | T1 | T1 | u | s | H | u | s | H | u | s | H |
| `I2D Global` | $s$ | 69.4 | 37.2 | 37.2 | 59.1 | 79.7 | 67.8 | 28.5 | **59.1** | 38.4 | 28.4 | 88.2 | 43.0 |
| `I2D Attention` | $s_{local}$ | 63.0 | 34.7 | 30.3 | 54.5 | 79.5 | 64.6 | 24.9 | 51.5 | 33.5 | 22.3 | **92.3** | 35.9 |
| **I2DFormer** | $s_{local}$ | 73.1 | 43.6 | 37.6 | 62.1 | **80.1** | 70.0 | 32.8 | 52.0 | 40.3 | 35.5 | 77.8 | 48.8 |
| **I2DFormer** | $s$ | **76.4** | **45.4** | **40.0** | **66.8** | 76.8 | **71.5** | **35.3** | 57.6 | **43.8** | **35.8** | 91.9 | **51.5** |

Table 1: **Further Ablation for I2DFormer** on the I2DGlobal and I2DAttention modules. I2DGlobal computes the global score $s$ and I2DAttention computes the local score $s_{local}$. We observe that jointly training both leads to the best performance for both $s_{local}$ and $s$ setting a SOTA. In ZSL, we report top-1 accuracy (**T1**) on unseen classes, in GZSL on seen/unseen (**s/u**) classes and their harmonic mean (**H**).

|  |  | Zero-Shot Learning | | | Generalized Zero-Shot Learning | | | | | | | | |
| --- | --- | --- | --- | --- | --- | --- | --- | --- | --- | --- | --- | --- | --- |
|  |  | AWA2 | CUB | FLO | AWA2 | | | CUB | | | FLO | | |
|  | Model | T1 | T1 | T1 | u | s | H | u | s | H | u | s | H |
| a) | `I2D Global` | 69.4 | 37.2 | 37.2 | 59.1 | 79.7 | 67.8 | 28.5 | **59.1** | 38.4 | 28.4 | 88.2 | 43.0 |
| b) | `I2D Global + D2I` | 67.1 | 39.5 | 32.0 | 53.9 | 76.5 | 63.2 | 32.0 | 61.4 | 42.1 | 28.3 | 87.0 | 42.7 |
| c) | `I2DFormer + D2I` | 68.7 | 42.5 | 37.6 | 58.1 | 76.3 | 66.0 | 32.3 | 52.8 | 40.1 | 34.2 | 86.0 | 48.9 |
| d) | **I2DFormer** | **76.4** | **45.4** | **40.0** | **66.8** | 76.8 | **71.5** | **35.3** | 57.6 | **43.8** | **35.8** | 91.9 | **51.5** |

Table 2: **Direction of Cross-Modal Attention.** We additionally ablate over Document to Image(D2I) attention as a counter to the proposed Image to Document(I2D) Attention in the main paper. While a document can describe the discriminative attributes of an image in addition to containing other non-visual information, an image only contains partial information about a document. We see that D2I attention suffers from this information asymmetry of the zero-shot setup and as a consequence, does not improve the performance. In ZSL, we report top-1 accuracy (**T1**) on unseen classes, in GZSL on seen/unseen (**s/u**) classes and their harmonic mean (**H**).

model) with D2I Attention to learn symmetric attention from Image to Document and from Document to Image. d) is our proposed model in the manuscript

**Results.** We see from the Table 2 that I2DFormer outperforms all baselines across the three datasets. As we compare rows a) and b), the introduction of the D2IAttention module leads to a drop in performance across two datasets due to the information asymmetry in the problem setting as discussed earlier. Row c) improves upon row b) as the model now additionally utilizes our I2DAttention module but its performance is limited by the D2IAttention. Our novel I2DFormer which only utilizes the I2DAttention module, outperforms all these baselines in row d). Our model is designed with the problem constraints of our ZSL setting and the resulting information asymmetry in mind. While being conceptually simple, it leads to significant performance gains as shown.

## 2 Example of documents.

We include two examples of documents for each dataset used as the auxiliary information by all methods in the main paper. The class names are hyperlinked to the original document source. We will release the full document set after the review process.

### 2.1 AWA2.

**Giraffe.**The Giraffe is an animal with an enormously long neck which allows it to exploit the leaves and vegetation that are too high up for other animals to find. Despite their length, the neck of the Giraffe actually contains the same number of bones as numerous other hoofed mammals but they are simply longer in shape. The Giraffe's elongated neck leads into a short body, with long and thin, straight legs and a long tail that is tipped with a black tuft that helps to keep flies away. The Giraffe tends to be white in colour with brown or reddish markings that cover its body (with the exception of their white lower legs). The markings of each Giraffe are not only unique to that individual but they

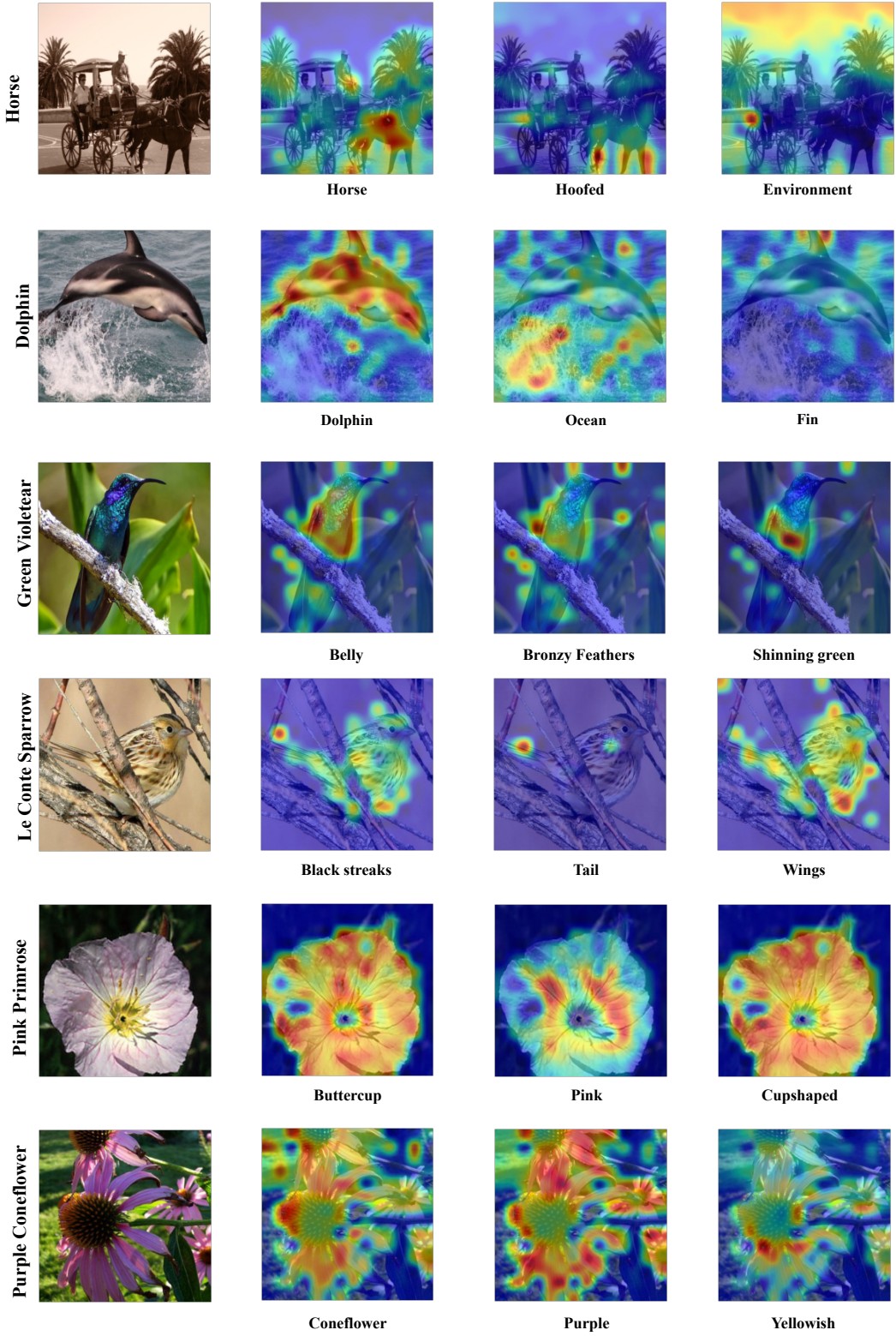

Figure 1: **Visualizing Word to Image Attention** in the most attended document words for `I2DEmb`, we see that our model I2DFormer has learned to localize them in the image without any paired patch-word supervision.

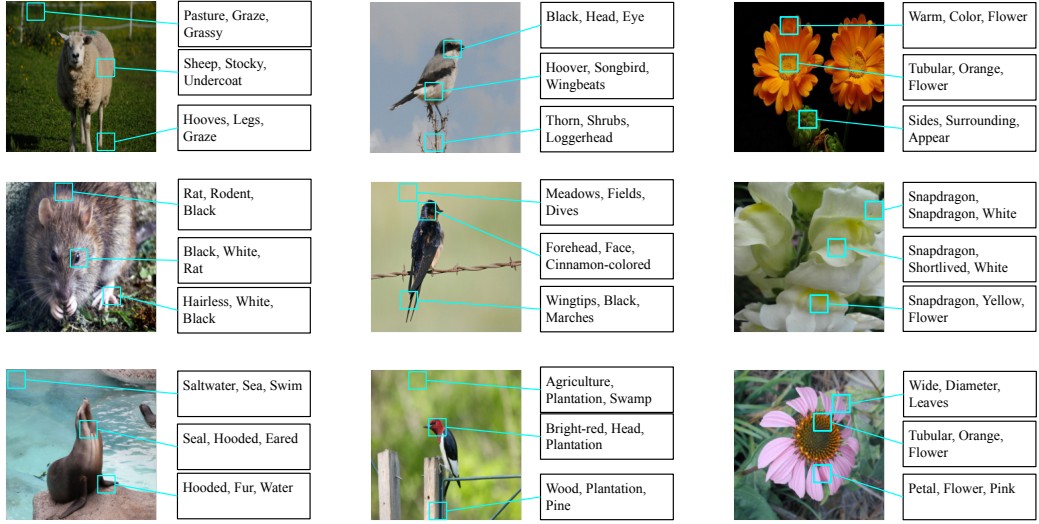

Figure 2: **Visualizing Image to Document Attention** we see that the top3 words contributing to each patch are visually grounded in the image.

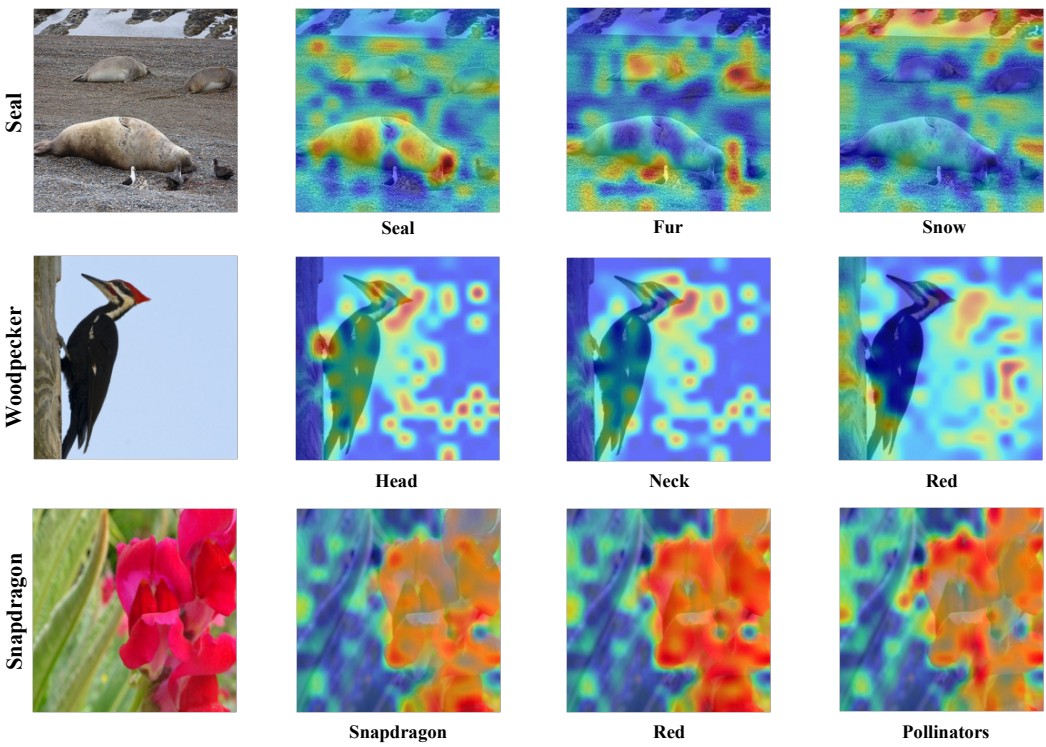

Figure 3: **Limitation of Image to Document Attention.** We see that while our model I2DFormer grounds document words in images, it comes with some limitations. Namely, the grounding is less accurate for especially for instances where unseen images vary significantly from the training data. However it is important to note that the model uses no paired supervision between image regions and document words but rather discovers them automatically using our attention mechanism with only image-level labels.

| | Classname | Top attended words for `I2DEmb` |
|---|---|---|
| **AWA2** | **Horse** | black, hoofed, horse, leg, feet, necks, tails, environment |
| | **Bat** | gray, hanging, bat, wings, feet, wing, brown, black |
| | **Rat** | gray, rodent, tail, black, sharp teeth, fur, incisors |
| | **Bobcat** | mountain, lynx, tail, bobcat, black markings, coat, mottled, camouflaged |
| | **Dolphin** | dolphin, ocean, fin, pods, snouts, flaps, blubber, bulbous |
| **CUB** | **Yellow head Blackbird** | yellow, chests, duller, yellow-headed, blackbirds, white patches, cattails, grackle |
| | **Le Conte Sparrow** | black streaks, tail, wings, orange, sparrow, meadows, breast, neck |
| | **Barn Swallow** | blue, crown, cinnamon-colored, forehead, meadows, tawny, white, tawny |
| | **Caspian Tern** | seabird, crown, plunges, black, ring-billed, white, speckling |
| | **Blue winged Warbler** | warbler, blue-winged, brushy, undertail, green, flocks, pointy, eyeline |
| **FLO** | **Pink Primrose** | buttercup, pink, cupshaped, fragrant, ranunculus, primrose, four-petaled, fused |
| | **Purple Coneflower** | coneflower, purple, yellowish, purpurea, protuberance, cone-shaped, hermaphrodite |
| | **Spear Thistle** | pink-purple, spear-shaped, monocarpic, gray, spined, feathery, thistle, weed |
| | **Monkshood** | Blue, poisons, lavendar, monkshood, blue-purple, aconitum, crowned, trilobed |
| | **Snapdragon** | lavender, snapdragon, mediterranean, lance-shaped, pollinators, symmetrical, opens, short-lived |

Table 3: **Top attended words for `I2DEmb` for unseen classes** in the Document Transformer consist of discriminative properties available in the document.

also vary greatly between the different Giraffe species in size, colour and the amount of white that surrounds them. All Giraffes though have large eyes that along with their height give them excellent vision, and small horn-like ossicones on the top of their heads. Giraffes are animals that inhabit open woodlands and savannah where using their height they are able to see for great distances around them to watch out for approaching danger.

**Horse.** The size and weight of these animals vary greatly from one breed to another. However, they all have the same general characteristics. The height is measured in hands instead of in inches. One hand equals the same as about 10cm or 4 inches. All horses have long necks that hold up their large, long heads. They have big eyes and ears, which are well-adapted for many environments. A mane of long hair grows down along their necks and their short tails are covered in coarse hairs, too. They come in a variety of colors because they have been bred so long for different traits. These animals are famously a hoofed mammal with one large 'toe' at the end of each leg. Their hooves consist of horn material which comes in different colors. Black is the most common hoof color, but horses with white feet often have white hoofs. White hooves are actually more brittle than pigmented ones. Appaloosa horses have a beautiful mixture of multiple colors. These types of painted horses often have striped hoofs that include both pigmented and white hoof material. These animals are well-suited to all kinds of environments and climates. Domestic horses can live almost anywhere as long as they have shelter, food, and space to run.

## 2.2 CUB.

**Cape May Warbler.** A delicate, short-tailed warbler with a slender and distinctively decurved bill, unusual among warblers. Larger than a Ruby-crowned Kinglet, smaller than a Song Sparrow. Both Sexes Length: 4.7-5.1 in (12-13 cm) Weight: 0.4-0.5 oz (10.2-15.2 g) Wingspan: 7.9-8.7 in (20-22 cm). Adult males are rich yellowish olive above, with rufous cheeks (auriculars) framed in yellow and dense rufous "tiger stripes" on the breast (present in all plumages, giving the species its scientific name, tigrina). The underparts are yellow, and the wing has a large white patch. Females and immatures are duller, lacking vivid yellow but with a yellowish green rump. Cape May Warblers hunt insects among branches, sip nectar from flowers, or eat fruit. They take most food by probing and picking but also catch insects in midair or hover to pluck items from leaves and branches. Boreal forest (spruce, balsam fir) in breeding season; a wide variety of forested and shrubby habitats during migration and in winter.

**Tropical Kingbird.** A powerfully built flycatcher with a big head and a heavy, long bill. It has pointed but broad wings, and its medium-length tail shows a shallow notch in the center. Larger than a Western Kingbird, smaller than a Green Jay. Both Sexes Length: 7.1-9.1 in (18-23 cm) Weight: 1.1-1.5 oz (32-43 g). A gray-headed bird with bright yellow underparts and a pale gray-green back. It has a whitish throat and dark gray-brown wings and tail. Forages by catching large flying insects on the wing, sallying out from a favored perch (often a telephone line) and returning to it to consume

the prey. Also feeds on fruits, particularly during cooler weather. Perches conspicuously and when nesting flies out to confront many sorts of birds that come too close to the nest. Found in almost any open or semiopen habitat in most of the range; in the United States, favors parks, towns, and rural areas with scattered trees for nesting and other perches for hunting, often near water. Ornithologists recognize three subspecies, which differ mostly in intensity of their plumage colors: satrapa from the United States south to Venezuela; despotes in eastern Brazil; and the larger melancholicus across the remainder of South America. Both despotes and melancholicus are more vividly yellow below than satrapa.

## 2.3 FLO.

**Peruvian Lily.** Alstroemeria, commonly called the Peruvian lily or lily of the Incas, is a genus of flowering plants in the family Alstroemeriaceae. They are all native to South America although some have become naturalized in the United States, Mexico, Australia, New Zealand, Madeira and the Canary Islands. Plants of this genus grow from a cluster of tubers. They send up fertile and sterile stems, the fertile stems of some species reaching 1.5 meters in height. The leaves are alternately arranged and resupinate, twisted on the petioles so that the undersides face up. The leaves are variable in shape and the blades have smooth edges. The flowers are solitary or borne in umbels. The flower has six petals each up to 5 centimeters long. They come in many shades of red, orange, purple, green, and white, flecked and striped and streaked with darker colors. There are six curving stamens. The stigma has three lobes. The fruit is a capsule with three valves. Alstroemeria are classified as an inferior monocot, meaning the petals are located above the ovary and the leaf veins are parallel.

**Globe Thistle.** Echinops is a genus of about 120 species of flowering plants in the daisy family Asteraceae, commonly known as globe thistles. They have spiny foliage and produce blue or white spherical flower heads. They are native to Europe, east to central Asia, and south to the mountains of tropical Africa. Globe thistle is the host plant of weevil Larinus vulpes. Echinops sphaerocephalus is a glandular, woolly perennial herbaceous plant with an average height of 50–100 centimetres (20–39 in), occasionally reaching 200 cm (80 inches). Its erect branching, gray, slightly wrinkled and hairy stems bear the occasional large, soft, sharply toothed, sharp-lobed pointed green leaves. They are sticky hairy above, and white woolly below. Atop each stem is an almost perfectly spherical inflorescence up to 6 cm in diameter, packed with white or blue-gray disc florets. It flowers from June until September. The flowers are pollinated by insects (usually bees, wasps and butterflies) (entomogamy) and are hermaphrodite (self fertilization or autogamy). The fruits are hairy cylindrical achenes about 7 to 8 mm long. They ripen from September through October. The seeds are dispersed by wind (anemochory). This species is widespread across much of Eurasia but it lives on other continents where it was introduced, including North America where it is a widespread weed.

## 3  Training Details.

We implement our model in PyTorch and train on an Nvidia A100 GPU. We use the VIT/B16 checkpoint trained on ImageNet 1k by [1] as the pretrained Image Transformer. This checkpoint respects the GBU split[5] i.e. no unseen class was used in pretraining of the feature extractor. Our patch projection and token projection layers are implemented as shallow MLP consisting of linear layers, followed by LayerNorm, ReLU, and Dropout. Hyperparameters are chosen based on the compute budget of training on a single A100 GPU and the best performance on the validation set. We hope that the ability to train I2DFormer on a single GPU enables its reproducibility in academic labs. We use a batch size of 64 for all three datasets. The method converges to the reported numbers within 24 hours. For the fine-grained datasets, we observe that weighing $L_{CLS}$, $L_{local}$ equally leads to the best performance since these are fine-grained datasets and benefit greatly from the I2D Attention. The source code and the reported trained checkpoints will be released after the review process. The detailed hyperparameter information of experiments reported for I2DFormer in the main paper is as follows.

**AWA2.** Image Patch projection: 3 layer MLP, Token projection layer: 2 layer MLP, Document Transformer: 2 encoder blocks with 4 multiheaded attention heads. Joint Embedding space size $r$: 256. Lambda $L_{CLS}$: 0.9, $L_{local}$: 0.1. Global average pooling: maxpool.

**CUB.** Image Patch projection: 2 layer MLP, Token projection layer: 2 layer MLP, Document Transformer: 2 encoder blocks with 4 multiheaded attention heads. Joint Embedding space size $r$: 64. Lambda $L_{CLS}$: 0.5, $L_{local}$: 0.5. Global average pooling: meanpool.

**FLO.** Image Patch projection: 3 layer MLP, Token projection layer: 2 layer MLP, Document Transformer: 2 encoder blocks with 4 multiheaded attention heads. Joint Embedding space size $r$: 128. Lambda $L_{CLS}$: 0.5, $L_{local}$: 0.5. Global average pooling: maxpool.

### 3.1 Open source artifacts.

The following open source artifacts were used in our experiments. We want to thank the respective authors for maintaining an easy-to-use codebase.

- APN
- GAZSL
- f-VAEGAN-D2
- VGSE
- MPNet
- LongFormer
- TFIDF
- X-Clip
- VIT
- ViLBERT