# OpenReview forum: "I2DFormer: Learning Image to Document Attention for Zero-Shot Image Classification"
_NeurIPS.cc/2022/Conference — NeurIPS 2022 Accept_

### Official Review · Reviewer_uvjr · 2022-07-06

**Rating:** 3
**Confidence:** 3
**Soundness:** 2 fair
**Presentation:** 1 poor
**Contribution:** 2 fair

**Summary:**

The paper proposes to use multimodal cross-attention to align image representation with word representation by unsupervised training on image-document pairs. The model is later validated on a Zero-Shot Learning task, and an interpretability study is conducted.


**Questions:**

See Cons in the previous section, and please clarify:
1. how the training proceeds,
2. how datasets are constructed (especially where the class information comes from),
3. what is the rationale for processing text with GloVe before passing it to Transformer?

**Limitations:**

The authors partially analyze the results and provide some details on when the model fails.

**Strengths And Weaknesses:**

Pros:
1. The interpretability study demonstrates that the model learned to align modalities together.
2. The performance over the baselines was improved.
3. The idea of Image2Document attention seems natural - using the asymmetric attention(image-query) on text-image pairs and aligning them into the same embedding space.

Cons:
1. Motivation: task description is short and appears relatively late in the paper(Zero-Shot Learning is a very vague statement). It is hard to assume what real-life problem the paper is trying to solve, which requires considerable mental effort to judge the model correctly. For example, the article's structure makes it impossible to understand the contribution while reading for the first time (not enough context on them is provided before). Both those issues make it hard to understand the significance of the contribution.
2. Training: It remains unclear how the training proceeds, e.g., how is the output of the similarity function classified? How are the images aligned to the text? Where does the class information come from?
3. Model: As I understand, the texts(words) are processed by converting them into GloVe embeddings, then through the MLP layer to be consumed by a Transformer network. This approach added a couple of processing steps, but the decision to do so remained unexplained and is not grounded in the experimental results anywhere in the paper. How much did using GloVe&MLP help? Additionally, in some aspects, the usefulness of this method counters the conclusion the paper reaches, especially the following: "Documents of unseen classes use the same and additional vocabulary in new sentences causing a distribution shift in their input representation." This issue has been previously tackled in other areas of NLP with the subword tokenization (e.g., BPE). Why the authors use static word embeddings in such a situation remains highly mysterious.
4. Model: Is there any reason not to try to unfreeze the pretrained visual encoder?
5. Model: The proposed model trains a single attention layer while the ablation on scaling the model deeper is lacking but may be significant.
6. Model: The choice of asymmetric attention(image-query) is not explained nor studied - one can imagine using the coattention or merging the inputs before passing to the self-attention (see, e.g. [1] for comparison). Is there any specific reason to design it that way?
7. Clarity: The caption in Table 1 is not clear: how were semantic embeddings learned from different sources, and were they represented?
8. Clarity: The paper presentation is aesthetically pleasing, but the structure makes it hard to understand and stack all the concepts together. Sadly, after reading the paper and having it in front of me, I would not be able to reproduce and reimplement the model's training.

---

> ### Author Response · Authors · 2022-08-02
> **Thank you for the helpful review! (Response to Reviewer uvjr 1/5)**
>
> We want to thank the reviewer for their review. The reviewer appreciated the interpretability of the model, the performance gains, and the novelty of our I2DAttention Module. The reviewer raised some concerns, which we address individually below. Please note that references to the manuscript and the supplementary e.g. line numbers/ sections are for the updated version for the rebuttal.
>
> ### **W1, W8) task description is short and appears relatively late, structure makes it hard to understand, what real-life problem the paper is trying to solve**
>
> Our paper follows the same format as previously published zero-shot learning papers at NeurIPS including [60, D]. These works use Zero-Shot Learning, synonymous with Zero-shot image classification. In this line of work, the core problem we and other works are trying to solve is classifying images of unseen classes that were not observed during training using side information. Our manuscript title mentions that the work is addressing zero-shot image classification. Although remarkable progress has been made towards zero-shot image classification, most prior works rely on human annotated attributes as the side information. Towards unsupervised semantic embeddings, word embeddings can be easily obtained from pre-trained language models. Yet, they often do not reflect fine-grained visual similarities, thus limiting the performance. The goal of this work is to learn visually aligned unsupervised semantic embeddings from online textual documents for zero-shot image classification. Towards this our proposed I2DFormer utilizes free form textual description to learn zero-shot classification and achieves SOTA on three benchmark datasets. Moreover, the learned class embeddings of our method can improve existing ZSL methods as shown in Table 2.
>
> We thank the reviewer for the suggestions regarding increasing clarity for a wider set of audience. We have modified the introduction section to highlight further the task description, problem setting, and contributions in the updated draft. We want to highlight that all the three other reviewers have rated our manuscript very high for presentation with a score of 3. We hope these modifications address the reviewer's concern.
>
> ### **W2, Q1) how the training proceeds?**
>
> We have provided a detailed method overview in manuscript section 3 with additional training details in supplementary Section 3. In addition to these, we provide details of our training and inference pipelines below for the reviewer. We hope this helps the reviewer get an overview.
>
>
>
> ***For Training:***
>
> **Input to the model:** an image and the set of documents for the seen classes.
>
> Step 1: Input the image and documents to the respective transformer to get the feature representation for the global CLS tokens, image patches and document tokens.
>
> Step 2.1: The dot product of the image CLS token and each Document CLS token is used in Equation 1 to define class scores “s(x, d)” using the document information per class.
>
> Step 2.2: The image patch and text token features are processed by the learnable attention module defined in section 3.2. Equation 3 computes an attention matrix between image patch queries and document token keys for a given image and each seen document. This attention is used to recompute the image patch embeddings f_{pa} from document token values which are pooled to give an image level feature embedding \hat{f}_{pa} used to generate the “s_local(x, d)” in equation 4 for each seen class.
>
> Step 3: The global class scores “s(x, d)” are optimized using the L_{CLS} which is a cross entropy over the set of documents for seen classes in Equation 2. Similarly the local class scores “s_local(x, d)” is optimized in L_{local} as a cross-entropy over the set of documents for seen classes in Equation 5.
>
> Step 4: The gradients from both the loss functions are used to update the model using Adam Optimizer
>
> This is repeated until training converges.
>
>
>
> ***At inference:***
>
> **Input to the model:** an image and the set of documents for both seen and unseen classes.
>
> Step 1: Input the image and documents to the respective transformer block to get the feature representation for the global CLS tokens of image and documents.
>
> Step 2: The dot product of the image CLS token and each Document CLS token is used in Equation 1 to define class scores “s(x, d)” using the document information per class.
>
> Step 3: An argmax over the scores for the given image and the documents for all classes is used to get the class output in Equation 6
>
> Moreover, we want to add that we will release the code and pre-trained models to promote reproducibility once the paper is accepted.
>
>
> ### **W2) Where does the class information comes from?**
> We formulate the problem setting in the beginning of Section 3 (line 99). The class labels of images are provided by the benchmark datasets. Each class is associated with exactly one textual document.

---

> > ### Author Response · Authors · 2022-08-02
> > **(Response to Reviewer uvjr 2/5)**
> >
> > ### **W3.1, Q3) the texts(words) are processed by converting them into GloVe embeddings, then through the MLP layer, the decision to do so remains unexplained, usefulness of this method counters the conclusion the paper reaches.**
> > Since we only have limited text in the form of documents of seen classes while training, we can not learn input features for all vocabulary from scratch as our model will observe new concepts at test time. It is critical to initialize the embeddings with a model which can represent all concepts of the language the model will encounter during training for seen classes and during testing for both seen and unseen classes. It is common practice in ZSL literature to improve upon this initial representation of a language model (for class names in their case) by refining it by various learnable models e.g. by GCN[29], Generative model[58], MLP[26] or a Transformer in our case. The intuition here is that the learned ZSL model improves the semantic space of concepts in relation to the same input language space which can represent both seen and unseen classes. To address the reviewer's request and show the usefulness of the MLP, we report an additional ablation in the table below. We compare using fixed GloVe vectors as input to the document transformer vs learning an MLP on top. We see that our strategy of learning an MLP on top of GloVe vectors greatly benefits the learnable document transformer.
> >
> > |    | **Doc Embedding** | **ZSL**  |          |          | **GZSL** |          |          |          |          |          |          |          |          |
> > |----|-------------------|----------|----------|----------|----------|----------|----------|----------|----------|----------|----------|----------|----------|
> > |    |                   | **AWA2** | **CUB**  | **FLO**  |          | **AWA2** |          |          | **CUB**  |          |          | **FLO**  |          |
> > |    |                   |          |          |          | **u**    | **s**    | **H**    | **u**    | **s**    | **H**    | **u**    | **s**    | **H**    |
> > | a) | GloVe             | 66.8     | 39.7     | 32.0     | 60.3     | 74.9     | 69.9     | 31.7     | 54.8     | 40.2     | 30.0     | 86.8     | 44.5     |
> > | b) | GloVe + MLP       | **76.4** | **45.4** | **40.0** | **66.8** | **76.8** | **71.5** | **35.3** | **57.6** | **43.8** | **35.8** | **91.9** | **51.5** |
> >
> > Our results do not contradict our claim as the utility of the word embedding model is to represent all concepts in the same semantic space and then learn an improved representation on top as it is common in a plethora of ZSL works including [29, 58, 26]. The point made in the paper for Table 3b) refers to context conditioned embeddings generated by the pretrained transformer based language models used as input to our learnable document transformer. To reiterate this, for example let's take the following two sentences “a horse has hoofed legs” and “a giraffe has long legs”. LongFormer will give different token embeddings for the token “legs”, while GloVe will embed the two instances of “legs” to the same initial embedding. For the input embedding from the LongFormer, the ZSL model will be faced with a distribution shift for the concept of leg as it has not observed enough variation of the token “legs” while training to generalize to this context changing input embedding.

---

> > > ### Author Response · Authors · 2022-08-02
> > > **(Response to Reviewer uvjr 3/5)**
> > >
> > > ### **W3.2) using subword tokenizer, using static embeddings**
> > > We also want to clarify a potential misunderstanding. Our embeddings are not static, only the initial embedding layer of the document transformer is static. These embeddings are refined by the learnable MLP and then further improved by the learnable Document Transformer. The subword tokenizers referenced by the reviewer are already part of baseline embedding models in table 3b), and it tackles a different problem of embedding rarer language terms as a combination of their simpler more frequently occurring sub-words. This compositionality from subwords is achieved by pre-training on Billions of Language data points to cover enough vocabulary. This can not be used to learn a semantic space over e.g. animal classes from just 40 documents of average length of 400 words each. The new information, especially nouns and adjectives detailed in the unseen documents e.g. hoofed legs, or the classname horse can not be sufficiently represented in the embedding space by sub tokenizing since nouns in language are not compositional with sub token configurations in this data scarce setting. All language embedding models use Billion scale training data including GloVe/ Word2Vec(Common Crawl of 840Billion tokens), Bert (Corpos of 3.3 Billion Tokens) etc. In addition aligning vision and language modalities with fully learnable embeddings is even more data hungry where works like CLIP[37] use 400 million images with captions and even works which use pretraining in the language domain still require 88 million image captions pairs to be competitive[A]. These modern transformer based pretrained embeddings are still not superior to classic word embeddings as shown in our work (also appreciated by reviewer oTDz). A similar conclusion is reached in [B], where authors showed that simple embedding models like FastText achieve comparable performance with CLIP and other Transformer based embeddings for language guided metric learning. We see this as an important result to motivate the community to further develop better embedding methods for zero-shot learning.
> > >
> > >
> > >
> > > ### **W4) Is there any reason not to try to unfreeze the pretrained visual encoder?**
> > > We fix the Image Transformer as pre-trained on ImageNet1K (which respects the GBU split i.e. the pretraining does not include any data of the unseen classes). We learn the Document Transformer from scratch. The choice of this training strategy is motivated by several published works including [66,C]. Namely in [66], the authors have studied the impact of freezing, finetuning and end to end training of image-text models with noisy text for the zero-shot transfer models like CLIP. The conclusion from these works propose that when dealing with noisy text, it is better to pretrain the image encoder and learn the text encoder from scratch to align the text modality to a common semantic space. Since our training setting has even more noise in the text domain in the form of non-visual facts about the class described in the document, we found this training strategy to perform the best for us. Moreover, it is important to know that we are working in a data-scarce setting where we only have one document for each class, and a few thousand training images in total.
> > >
> > > ### **W5, 6) scaling the model deeper is lacking but may be significant, the choice of asymmetric attention**
> > > We want to clarify that since both of our image and document modalities are processed by respective transformers, these already contain several self attention layers. However, we only use one cross-modal attention layer in the final model. We touch on scaling up cross-modal attention in section 4.3 and Table 3a) in comparison with ViLBERT (line 293-296). Specifically, in our data constraint setup, deeper cross-modal attention configurations lead to suboptimal performance and a parameter efficient model like ours leads to better performance (as also appreciated by Reviewer gQFr).

---

> > > > ### Author Response · Authors · 2022-08-02
> > > > **(Response to Reviewer uvjr 4/5)**
> > > >
> > > > [Continued for W5, 6]
> > > >
> > > > We want to thank the reviewer for raising this important point. We have updated the supplementary to discuss this problem constraint in detail in Supplementary Section 1.4 to empirically confirm our design choices. The attention direction of our model is motivated by the information asymmetry in our problem setting. A document describes the most discriminative information in the image along with non-visual information about the class. Our I2DAttention learns to focus on the visual information to align the two modalities while learning to limit the impact of non-visual information. An image, however, only contains limited information about the document and does not contain features about the non-visual content of the document. As a consequence, learning Document to Image attention can lead to picking up spurious correlations which can limit the model’s performance. We would like to point out that this limitation originates from the ZSL nature of our problem, as we only have one textual document to represent each class in contrast to zero-shot transfer works like CLIP[37] where a caption locally describes the image for each training iteration.
> > > >
> > > > We have included additional ablation about Document to Image attention, Symmetric attention and deeper attention configuration in the table below. We show that our proposed model achieves the best performance.
> > > >
> > > >
> > > >
> > > > ***Experimental Setup:***
> > > >
> > > > Baseline a) I+D TokenFormer concatenates the image and article tokens and inputs this sequence into a learnable transformer Encoder block to have full attention between each modality and cross-modality. The transformer encoder block outputs a CLS token which is used for classification.
> > > >
> > > > Baseline b) is our I2DGlobal module introduced in Section 3.1.
> > > >
> > > > Baseline (c-e) introduces the D2IAttention module which is the symmetric counterpart to our I2DAttention module for Document to Image Attention.
> > > >
> > > > Baseline c) combines I2DGlobal with D2IAttention similar to our proposed model to learn asymmetric attention from Documents to Image.
> > > >
> > > > Baseline d) combines I2DFormer(the proposed model) with D2IAttention to learn symmetric attention from Image to Document and from Document to Image.
> > > >
> > > > Baseline e) combines the I2DGlobal with I2DAttention and D2IAttention. However, instead of pooling the token dimension to compute class scores (Equation 4 of the manuscript), we concatenated the attention recomputed image and document token embeddings and pass them through a Transformer encoder block similar to a) for an additional full attention layer between the two modalities to compute the class score. This baseline is included to study if scaling up cross modal attention will improve performance.
> > > >
> > > > f) is our proposed model I2DFormer from the manuscript
> > > >
> > > > |    | **Model**            | **ZSL**  |          |          | **GZSL** |          |          |          |          |          |          |          |          |
> > > > |----|----------------------|----------|----------|----------|----------|----------|----------|----------|----------|----------|----------|----------|----------|
> > > > |    |                      | **AWA2** | **CUB**  | **FLO**  |          | **AWA2** |          |          | **CUB**  |          |          | **FLO**  |          |
> > > > |    |                      |          |          |          | **u**    | **s**    | **H**    | **u**    | **s**    | **H**    | **u**    | **s**    | **H**    |
> > > > | a) | I+D TokenFormer      | 66.8     | 34.1     | 29.7     | 59.1     | 72.2     | 65.0     | 26.5     | 47.4     | 33.9     | 26.7     | 87.3     | 40.9     |
> > > > | b) | I2D Global           | 69.4     | 37.2     | 37.2     | 59.1     | **79.7** | 67.8     | 28.5     | 59.1     | 38.4     | 28.4     | 88.2     | 43.0     |
> > > > | c) | I2D Global + D2I     | 67.1     | 39.5     | 32.0     | 53.9     | 76.5     | 63.2     | 32.0     | **61.4** | 42.1     | 28.3     | 87.0     | 42.7     |
> > > > | d) | I2DFormer + D2I      | 68.7     | 42.5     | 37.6     | 58.1     | 76.3     | 66.0     | 32.3     | 52.8     | 40.1     | 34.2     | 86.0     | 48.9     |
> > > > | e) | I2DFormer + D2I + a) | 67.9     | 42.1     | 36.6     | 55.4     | 78.0     | 64.8     | 31.4     | 55.3     | 40.0     | 28.9     | 90.9     | 43.9     |
> > > > | f) | **I2DFormer(Ours)**  | **76.4** | **45.4** | **40.0** | **66.8** | 76.8     | **71.5** | **35.3** | 57.6     | **43.8** | **35.8** | **91.9** | **51.5** |

---

> > > > > ### Author Response · Authors · 2022-08-02
> > > > > **(Response to Reviewer uvjr 5/5)**
> > > > >
> > > > > [Continued from the last table]
> > > > >
> > > > > ***Results:***
> > > > >
> > > > > We see from the table that our I2DFormer (row f) outperforms all baselines across the three datasets. As we compare rows b) and c), we observe that the introduction of the D2IAttention module leads to a drop in performance across two datasets. We attribute the relatively low performance of I+D TokenFormer (row a) to the same issue of information asymmetry in addition to the extra learnable parameters introduced by the full Transformer Encoder. We would also like to point out that I+D TokenFormer also involves several memory repeats of image and document tokens to concatenate the two modalities which results in a 7x increase in required GPU memory. Comparing row e) and a), we see that the introduction of our Attention module improved the performance of I+D TokenFormer but the performance is limited by Document to Image Attention in addition to extra learnable parameters introduced by additional attention blocks. We want to highlight that in our data scarce ZSL classification setting, scaling up additional attention modules is not the optimal way to align the two modalities. Our novel I2DAttention module is designed with these problem constraints in mind. While being conceptually simple, it leads to significant performance gains as shown in row f).
> > > > >
> > > > >
> > > > >
> > > > > ### **W7) how were semantic embeddings learned from different sources**
> > > > >
> > > > > The experimental setup of the compared semantic embeddings has been described in details in section 4.1, “Compared Semantic Embeddings” at line 230. Namely, the embeddings are extracted by the recommended procedure from the works cited in Table 1. The details are omitted from the caption for brevity as it is the standard embedding procedure for these models and already described in Section 4.1.
> > > > >
> > > > >
> > > > >
> > > > > ### **Q2) how datasets are constructed**
> > > > >
> > > > > The three compared datasets, AWA2, CUB, and FLO are standard zero-shot learning datasets extensively studied in the literature[2, 58, 60, 68]. We have augmented these datasets with one text document for each class describing the respective class. The process of this document collection is described in Section 4, “Collecting documents” at line 203.
> > > > >
> > > > >
> > > > >
> > > > > ###  **References:**
> > > > >
> > > > > [A] “Supervision Exists Everywhere: A Data Efficient Contrastive Language-Image Pre-training Paradigm” Yangguang Li, Feng Liang, Lichen Zhao, Yufeng Cui, Wanli Ouyang, Jing Shao, Fengwei Yu, Junjie Yan, ICLR2022
> > > > >
> > > > > [B] “Integrating Language Guidance into Vision-based Deep Metric Learning” Karsten Roth, Oriol Vinyals, Zeynep Akata
> > > > >
> > > > > [C] “What Makes Training Multi-Modal Classification Networks Hard?” Weiyao Wang, Du Tran, Matt Feiszli, CVPR 2020.
> > > > >
> > > > > [D] “Generalized Zero-Shot Learning with Deep Calibration Network” Shichen Liu, Mingsheng Long, Jianmin Wang, Michael I. Jordan, NeurIPS 2018.

---

> ### Author Response · Authors · 2022-08-08
> **We thank and invite Reviewer uvjr to discuss our response.**
>
> Dear Reviewer uvjr
>
> We want to thank you once again for your helpful review. We have incorporated your feedback into the manuscript including additional discussion and experiments. We believe your suggestions further improved the clarity of the manuscript and opens it to a wider set of audience. We have also discussed your questions and concerns in detail in our rebuttal. Given we have less than two days left for author-reviewer discussion, we want to invite you to discuss our response.

---

### Official Review · Reviewer_Sfca · 2022-07-11

**Rating:** 6
**Confidence:** 4
**Soundness:** 3 good
**Presentation:** 3 good
**Contribution:** 3 good

**Summary:**

In this paper, authors propose to address the zero-shot image classification problem under a more realistic setting, namely each class is descried with one document collected online. To address the issues with online textual documents contain certain noise, and different parts of the document may correspond to the different regions of the images, authors propose a transformer-based ZSL framework that jointly learns to encode images and documents by aligning both modalities in a shared embedding space. The cross-modality attention mechanism is introduced to suppress the noise. And extensive experiments are conducted to validate the proposed modules.

**Questions:**

- The implementation details regarding the compared methods should be given, since different noisy textual source is utilized under the current experimental settings;

- What about the model complexity, FLOPs of the proposed models compared to other SOTA models?

- What if all tokens (both visual ones and semantic ones) are jointly fed to the transformer instead of utilizing the separate transformer then employing the cross-modal attention, since the transformer contains self and cross attention mechanism itself.

**Limitations:**

The main concern regarding the limitation is the model complexity.

**Strengths And Weaknesses:**

Strengths:
- The transform-based framework is proposed for the ZSL problem.

- The I2D module is proven effective to capture the relevant image regions based on the collected documents.

- The experimental results are considered favorable compared to SOTAs on three benchmark datasets. And the results are highly-interpretable.

- The learned text embedding can be utilized with existing ZSL for further improvement.

Weakness:
- The discussions regrading the model complexity needs further clarification.

- The novelty of cross modality attention among different tokens is somewhat limited being the main contribution of the whole paper.

- The fairness of utilizing the online document compared to other SOTA ZSL frameworks needs further clarification.

The more detailed comments I'd like authors to address are summarized in the Questions part.

---

> ### Author Response · Authors · 2022-08-02
> **Thank you for the encouraging and helpful review! (Response to Reviewer Sfca 1/2)**
>
>
>
> We thank the reviewer for the encouraging review and the helpful suggestions. The reviewer appreciated our transformer based model for ZSL learning, the interpretability of our model, the experimental evaluation and the improvement in baselines with our learned document embeddings. The reviewer also rated the paper high for soundness, presentation and contribution.
>
>
> We now address the individual comments. Please note that references to the manuscript and the supplementary e.g. line numbers/ sections are for the updated version for the rebuttal.
>
>
> ### **W1, Q2) The discussions regarding the model complexity needs further clarification, model complexity.**
>
> Since different ZSL and class embedding methods employ different training strategies including multi-step training e.g. in the case of VGSE[61], they are not directly comparable in FLOPs.
>
>
>
> Our method requires roughly the same compute and training time compared to our closest unsupervised class embedding competitor VGSE[61]. For comparison, our model is trained on a Single A100 GPU and requires only 24 hours to converge to the reported number while utilizing 20GB of VRAM(as also appreciated by Reviewer oTDz). In comparison VGSE takes a similar training time of 25 hours while utilizing 18GB of VRAM.
>
>
>
> Compared to baseline ZSL models like APN and f-VAEGAN-D2, our model requires relatively more training time and GPU memory. However, our model is learning both a ZSL model and a generic unsupervised class embedding that can be utilized by other methods. Once trained, our learned I2DEmb can benefit any of these baseline models as shown in Table 2.
>
>
> For model inference, our model’s computational requirements are very similar to the most basic baselines like SJE as we only require a dot product between each image CLS feature and the document CLS features which can be precomputed once for each evaluation run.
>
>
>
> ### **W2) Novelty of cross modal attention, other novelty**
>
> We want to highlight that our cross-modal attention module is fundamentally different from the existing published works compared in the manuscript Table 3a). Instead of learning deep or interleaved cross-modal attention layers, our I2DAttention module exploits the information asymmetry between the image and text domain to develop a parameter-efficient attention module for zero-shot image classification with text documents (as appreciated by Reviewer oTDz and gQFr). We have included additional discussion regarding this in supplementary section 1.4. In addition to our cross modal ablations in Table 3a) where we showed that our attention module outperforms existing published work, we address your suggestion of inputting all tokens to a transformer in the later comment. Our analysis shows that existing versions of attention will achieve suboptimal results due to not addressing the zero-shot nature of the problem. In addition to the novelty of our attention module, our model distills the knowledge of fine-grained attention to the global head to learn a computationally efficient inference model for test time. Our other contributions in this work include creating a document-based dataset for existing ZSL benchmarks, achieving SOTA performance on three public benchmarks, and analysis of augmenting existing word embedding and document embedding models with our transformer-based model.
>
>
>
> ### **W3) fairness of utilizing the online document compared to other SOTA ZSL frameworks**
>
> We follow the evaluation protocol of [61] where different unsupervised class embeddings are compared by replacing the original class embeddings of each ZSL method.  We believe that the comparison is fair as our experiments use the same input image features from a pretrained ImageNet1K model and the same documents. For a fair comparison with different unsupervised embedding methods, we have ablated over them under the same model and training protocol in Table 1. Additionally, we have ablated over several ZSL methods under different unsupervised class embeddings in Table 2 using the training protocol recommended by [61]. We conclude in the paper that our learned embeddings are superior to other unsupervised class embeddings in Table 1. Additionally we conclude that our model outperforms other baseline ZSL models across different class embeddings in Table 2. Finally our learned document embedding also leads to significant improvements in the performance of baseline ZSL methods as also shown in Table 2.
>
>
>
> ### **Q1) implementation details regarding the compared methods should be given, since different noisy textual source is utilized under the current experimental settings**
>
> We want to clarify that we use the same set of documents in both Table 1 and Table 2 as we compare different embedding methods and ZSL models. These embeddings are extracted using the respective author’s implementation. We discuss how each embedding is extracted in Section 4.1 at Line 230 “Compared semantic embeddings”.

---

> > ### Author Response · Authors · 2022-08-02
> > **(Response to Reviewer Sfca 2/2)**
> >
> > ### **Q3) What if all tokens (both visual ones and semantic ones) are jointly fed to the transformer?**
> >
> > We have included this baseline as I+D TokenFormer in the table below. We see from the table that I2DFormer outperforms this baseline across the three datasets. We attribute this to the information asymmetry between the image and document domain in our setting (discussed in more detail in supplementary section 1.4) and the increased learnable parameters of a full transformer encoder block for I+D TokenFormer. A document describes the most discriminative information in the image along with non-visual information about the class. Our I2DAttention learns to focus on the visual information to align the two modalities while learning to limit the impact of non-visual information. An image, however, only contains limited information about the document and does not contain features about the non-visual content of the document. As a consequence, learning additional Document to Image attention in this model can lead to picking up spurious correlations which can limit the model’s performance. We would also like to point out that I+D TokenFormer also involves several memory repeats of image and document tokens to concatenate the two modalities which results in a 7x increase in required GPU memory.
> >
> > |    | **Model**            | **ZSL**  |          |          | **GZSL** |          |          |          |         |          |          |          |          |
> > |----|----------------------|----------|----------|----------|----------|----------|----------|----------|---------|----------|----------|----------|----------|
> > |    |                      | **AWA2** | **CUB**  | **FLO**  |          | **AWA2** |          |          | **CUB** |          |          | **FLO**  |          |
> > |    |                      |          |          |          | **u**    | **s**    | **H**    | **u**    | **s**   | **H**    | **u**    | **s**    | **H**    |
> > | a) | I+D TokenFormer      | 66.8     | 34.1     | 29.7     | 59.1     | 72.2     | 65.0     | 26.5     | 47.4    | 33.9     | 26.7     | 87.3     | 40.9     |
> > | b) | **I2DFormer(Ours)**  | **76.4** | **45.4** | **40.0** | **66.8** | **76.8** | **71.5** | **35.3** | **57.6**    | **43.8** | **35.8** | **91.9** | **51.5** |

---

### Official Review · Reviewer_gQFr · 2022-07-11

**Rating:** 6
**Confidence:** 4
**Soundness:** 3 good
**Presentation:** 3 good
**Contribution:** 3 good

**Summary:**

The authors propose an attention-based model for zero-shot learning from text documents - knowledge sources such as Wikipedia and bird/flower information websites. The proposed approach relies on two Transformer models: one for processing image patches, and another one for text documents. Each of the Transformers outputs a sequence of features which is fed into two different components of the proposed model. One of them fuses two sequences with an attention mechanism (I2D Attention) and produces an image-text matching score, described as local. The other component (I2D Global) only uses special classification tokens ([CLS]) from both text and image and similarly produces an image-text matching score, which the authors describe as global. The approach optimizes for both local and global matching but only the global one is used later on for classification/generating predictions.

The authors evaluate their model on standard ZSL datasets: AWA2, CUB, and FLO. They separately evaluate the performance of both their learned text features and their entire model that uses the global matching scores. When using their learned text features as input to existing four different ZSL models they observe quite a consistent improvement over alternative GloVe or VGSE features, often by a big margin. Also, using their entire proposed model, with global text-image matching scores used for classification, generally compares well against the other ZSL models.

Additionally, the authors argue that their model is more interpretable, showing examples of attention maps over words in a document or matching image-text element pairs.

**Questions:**

**Questions:**

- (Q1) Table 3 (b): How are different input features (e.g. GloVe) used in the Document Transformer? Is it just an input to a Transformer instead of learning token embeddings (as in the first layer of Transformer)? Are those text features fixed or fine-tuned?

- (Q2) Are any of either the Image or Document Transformer fine-tuned? The Image Transformer seems to be mentioned to be pre-trained but what about Document Transformer? Is it also pre-trained in any way or trained from scratch?

- (Q3) What are the differences between text lengths of different documents? Since the softmax is computed over all classes, and texts lengths can be different, doesn’t it cause significant computational issues? Different lengths would require masking and padding

- (Q4) How does the proposed model compare with other ZSL models with respect to computational requirements - both in training and generating predictions?

- (Q5) How are soft attention maps produced (e.g. Figure 3) if the method only operates on image patches?

- (Q6) Analysis of differences in classification between $s$ and $s_{local}$ predictions (Table 1 in the Appendix): are there any important differences other than just a different level of accuracy? Do they make different types of mistakes? If the two modules are complementary to each other some error analysis from the corresponding scoring function could be insightful

- (Q7) Eq. 4: What is $\hat{x}\_{pa}$? Shouldn’t it be $\hat{f}_{pa}$ instead?
    - Why H in eq. 4 is a function while in L167 it says it’s a vector?

---

**Suggestions:**

- Ablations in the Appendix, Table 1 seem important - I would recommend at least mentioning the main observations in the main paper

**Minor issues:**

- L50: “consistent” —> “consistently”?

- Table 3: “Modal” —> “Model”?

- Figure 1 doesn’t seem to be referred to anywhere in the text

- Hyperlinks should be highlighted somehow - they do not seem to be displayed in any special way - just as normal text - very easy to miss them

- Terminology: Instead of document “embedding” - I’d rather say encoding/representations/features. Typically, “embedding” refers to basically a look-up for discrete objects (e.g. tokens) - the terminology used in this paper with “embeddings” used as a name for any features or representations makes it very easy to confuse with e.g. Transformer’s embeddings (in the very first layer).

**Limitations:**

The authors do not discuss the computational cost of training & evaluating their model as opposed to alternative approaches. The qualitative analysis seems to include only selected positive example, with no fail-case analysis (see W3). Additionally, the authors use some very strong, over-exaggerated language - in L45 they claim that their model is “able to develop understanding of different parts of an animal”.

**Strengths And Weaknesses:**

**Strengths:**

- (S1) The proposed approach is conceptually simple - it consists mostly of dot-product attention. It doesn’t require many extra hyperparameters.

- (S2) The experimental results. The learned document text features (I2DEmb) quite consistently, both among different datasets and different methods, outperform GloVe and VGSE features: often by a large margin. Additionally, the entire proposed model that uses a very simple similarity score between text and image [CLS] token often outperforms the remaining ZSL methods - even models that use the text features proposed in this work (I2DEmb)

- (S3) Rich ablation studies and a good set of experiments: the experiments show the importance of different components of the proposed model. The authors evaluate their model with different types of textual features but also use their textual features with different models. The set of experiments covers the most important aspects of the model.

- (S4) Attention maps provide some extra interpretability which can be important for understanding predictions, especially when using text documents as a source of information about classes (although see W3)

**Weaknesses:**

- (W1) Possibly suboptimal settings for ZSL models used to compare against. GAZSL was introduced using TF-IDF features, e.g. APN was introduced for attribute features. This paper however compares against them when using either GloVe or VGSE features - which might be suboptimal for those models. A more fair comparison should at least additionally include TF-IDF features as well (especially important for GAZSL).

- (W2) The authors seem to miss a little bit of context regarding the attribute data. The usage of text documents instead of attributes, in general, might have many advantages. However, e.g. f-VAEGAN-D2, APN papers using attribute data report much higher performance on e.g. CUB dataset. The authors however do not mention any attribute-based results which might be misleading for the readers, as if attribute features were not competitive even on these datasets. Additionally, the claims of outperforming SOTA should be formulated more precisely - e.g. with respect to a specific type of data/source of information

- (W3) The qualitative evaluation samples are not indicated as random samples, as opposed to selected. Additionally, for more convincing/insightful results an analysis of failure cases would be needed

- (W4) The authors, without justifying, collect their own text documents instead of already standard existing ones already extracted. There is no motivation for it present in the paper and no discussion or analysis of the differences and the impact of using these vs. standard documents. Additionally, the authors mention performing some filtering of document sections, which could potentially be very important but is not discussed/analyzed further
    - Existing extracted documents (Wikipedia, AllAboutBirds, etc.):
        - FLO, CUB: Mohamed Elhoseiny, Ahmed Elgammal, and Babak Saleh. Write a classifier: Predicting visual classifiers from unstructured text. TPAMI 2016.
        - CUB: Mohamed Elhoseiny, Yizhe Zhu, Han Zhang, and Ahmed Elgammal. Link the head to the” beak”: Zero shot learning from noisy text description at part precision. CVPR 2017

(W5) The “direction” of the attention seems somewhat arbitrary and not analyzed. The authors basically weight the text features (attention values) by the attention compatibility between image patches & text sequence elements. One could do the opposite - use image patch features and weight them by attention maps instead. Or do both directions - like some cross-attention works. No motivation behind this choice is discussed and no comparison of alternative choices is present.

---

> ### Author Response · Authors · 2022-08-02
> **Thank you for the very detailed and helpful review! (Response to Reviewer gQFr 1/4)**
>
> We want to thank the reviewer for the detailed and very helpful review. The reviewer appreciated the simplicity of our attention module and its parameter efficiency. The reviewer also appreciated the consistent SOTA performance of our model, rich ablation studies, and the extra interpretability offered by the attention module. The reviewer also rated the paper high for soundness, presentation, and contribution.
>
> We now address the mentioned comments. Please note that references to the manuscript and the supplementary e.g. line numbers/ sections are for the updated version for the rebuttal.
>
>
> ### **W1) Possibly suboptimal settings for ZSL models**
>
> We follow the evaluation protocol of VGSE [61] where different unsupervised class embeddings are compared by replacing the original class embeddings of each ZSL method.  In APN’s journal extension[A], the authors also compare the performance of unsupervised class embedding methods under the same setting. In the table below, we compare the performance of GAZSL with TF-IDF feature vs I2DEmb and show that our observations from the manuscript are consistent. Namely, GAZSL performs better with our learned document embeddings I2DEmb, and our I2DFormer still achieves the SOTA.
>
> |    | **Model**              | **ZSL**  |          |          | **GZSL** |          |          |          |         |          |          |          |          |
> |----|------------------------|----------|----------|----------|----------|----------|----------|----------|---------|----------|----------|----------|----------|
> |    |                        | **AWA2** | **CUB**  | **FLO**  |          | **AWA2** |          |          | **CUB** |          |          | **FLO**  |          |
> |    |                        |          |          |          | **u**    | **s**    | **H**    | **u**    | **s**   | **H**    | **u**    | **s**    | **H**    |
> | a) | GAZSL w. TF-IDF        | 48.0     | 39.2     | 33.1     | 28.0     | **95.2** | 43.3     | 9.58     | 54.2    | 16.3     | 27.1     | 91.5     | 41.8     |
> | b) | GAZSL w. I2DEmb (Ours) | **83.1** | 42.9     | 34.2     | 56.8     | 94.7     | 71.0     | 15.9     | 50.4    | 24.1     | 28.8     | 90.1     | 43.7     |
> | c) | **I2DFormer (Ours)**          | 76.4     | **45.4** | **40.0** | **66.8** | 76.8     | **71.5** | **35.3** | **57.6**   | **43.8** | **35.8** | **91.9** | **51.5** |
>
> ### **W2) claims of outperforming SOTA should be formulated more precisely - e.g. with respect to a specific type of data/source of information**
>
> We agree with the reviewer that our claims are with respect to unsupervised class embeddings and not human labeled attributes. We thank the reviewer for requesting additional clarity. We want to clarify that the focus of our work is to bridge the gap between ZSL performance using expensive human-annotated attributes vs cheap unsupervised class embeddings as mentioned in Intro line 30-33. We also mention this in our manuscript abstract(line 16-19), contribution point 3 where we state “Our model I2DFormer consistently improves the SOTA in unsupervised semantic embeddings”(line 50-52). Moreover, we start the section 4 of our manuscript as “Since the main focus of this work is to learn unsupervised semantic embeddings, we do not use any human-annotated attributes.”(line 198-200). As mentioned in the previous point, this protocol has been introduced by previously published work including [61]. To address the reviewer’s concern, we have added additional explanation in the intro (line 34-36) and have updated the caption of Table 1 and 2 to specifically mention that our claim of SOTA is wrt unsupervised class embeddings.
>
> ### **W3) qualitative evaluation samples are not indicated as random samples, failure cases.**
>
> The qualitatives included in the main manuscript were chosen for clarity. The samples in the supplementary were chosen randomly on correctly classified images. We agree with the reviewer that showing failure cases for attention will offer more insights. Therefore we have now included a discussion around failure cases in the updated supplementary section 1.3 and Figure 3 as requested by the reviewer. We show that since our model directly learns the attention from the data instead of paired supervision for image regions and document words, it is not immune to dataset biases. The learned attention can fail in cases where unseen classes have a large number of instances in a single image, significant orientation changes or the attribute of a flower varies significantly from the seen classes.
>
>
> ### **References:**
>
> [A] “Attribute Prototype Network for Any-Shot Learning” Wenjia Xu, Yongqin Xian, Jiuniu Wang, Bernt Schiele, Zeynep Akata, IJCV, 2022
>
> [B] “What Makes Training Multi-Modal Classification Networks Hard?” Weiyao Wang, Du Tran, Matt Feiszli, CVPR 2020.

---

> > ### Author Response · Authors · 2022-08-02
> > **(Response to Reviewer gQFr 2/4)**
> >
> > ### **W4) justifying collect their own text documents, performing some filtering of document sections**
> >
> > We want to thank the reviewer for citing the impactful related works. These works have also inspired our work and we have already cited them in our manuscript. The documents released by [14] are missing section information which prevented us from studying the impact of performing a relatively cheap section filtering step to reduce some noise in the collected document. Section filtering reduces the relative memory cost for attention as the document length is reduced from ~1500 words to ~400 words. Moreover, the mentioned works have collected their set of documents in 2016 and are therefore 6 years old. Since sources like Wikipedia are ever evolving with richer information, we felt that the effort spent in recollecting the documents can potentially benefit future works as we see a boom of good vision-text models thanks to Transformer architectures. For comparison, the authors mention in [14] that in 2016 by querying Wikipedia for classnames, they were able to get matches for 178/200 classes while in 2022, we were able to retrieve all 200/ 200 classes without manual intervention. We want to highlight that for CUB, we performed analysis over impact of document sources in supplementary 1.2 and show that our model is SOTA compared to baseline unsupervised embeddings on both Wikipedia and AllAboutBird documents.
> >
> > We also want to mention that we use the same set of newly collected documents for all baselines reported in the manuscript for a fair comparison. To address the reviewer’s request, we include additional ablation below on the impact of section filtering on Wikipedia articles for CUB. We chose Wikipedia documents for CUB for this study as they tend to contain more noise than the relatively cleaner AllAboutBirds reported in the main manuscript. The document collection protocol is the same as [14] i.e. we only query Wikipedia from the Python api and perform section filtering where required and train I2DFormer.
> >
> > |                        | **ZSL**  |          | **GZSL** |          |
> > |------------------------|----------|----------|----------|----------|
> > | **Input Document**     |          | **u**    | **s**    | **H**    |
> > | Only Abstract          | 40.3     | 31.7     | 52.6     | 39.5     |
> > | Full Article           | 42.5     | 33.1     | 56.2     | 41.7     |
> > | Visual Sections (Ours) | **43.1** | **34.1** | **57.1** | **42.1** |
> >
> >
> > We see from the table above that our strategy of filtering for Visual Sections achieves the best performance for both ZSL and GZSL. Row1 achieves decent results by performing a very simple filtering of only extracting the abstract of a document. However, as abstracts only contain partial information about the visual attributes of a class, it limits the model’s performance. Row 2, which uses the full article, improves upon this as the model now has access to additional class information. At the same time, this also increases the noise in the document due to its long length. However, our model is still able to learn a very competitive class embedding showing the robustness of our attention module. While competitive, using the full document has a disadvantage of requiring significantly more GPU compute to process the full document in the Document Transformer and subsequently the cross modal I2DAttention module. Finally, in row 3, we show that filtering for visual sections achieves the best performance as this simple step reduces the noise in the document in addition to reducing the required GPU compute.

---

> > > ### Author Response · Authors · 2022-08-02
> > > **(Response to Reviewer gQFr 3/4)**
> > >
> > > ### **W5) The “direction” of the attention seems somewhat arbitrary**
> > >
> > > We want to thank the reviewer for raising this important point. We have updated the supplementary to discuss this problem constraint in detail in Supplementary Section 1.4 to empirically confirm our design choices. The attention direction of our model is motivated by the information asymmetry in our problem setting. A document describes the most discriminative information in the image along with non-visual information about the class. Our I2DAttention learns to focus on the visual information to align the two modalities while learning to limit the impact of non-visual information. An image, however, only contains limited information about the document and does not contain features about the non-visual content of the document. As a consequence, learning Document to Image attention can lead to picking up spurious correlations which can limit the model’s performance.
> > >
> > >
> > >
> > > ***Experimental Setup:***
> > >
> > > Baseline a) is our I2DGlobal module introduced in Section 3.1.
> > >
> > > Baseline (b,c) introduces the Document to Image(D2I) Attention module which is the counterpart to our I2DAttention module.
> > >
> > > Baseline b) combines I2DGlobal with D2I Attention similar to our proposed model to learn asymmetric attention from Documents to Image.
> > >
> > > Baseline c) combines I2DFormer(the proposed model) with D2I Attention to learn symmetric attention from Image to Document and from Document to Image.
> > >
> > > d) is our proposed model in the manuscript
> > >
> > > |    | **Model**          | **ZSL**  |          |          | **GZSL** |          |          |          |          |          |          |          |          |
> > > |----|--------------------|----------|----------|----------|----------|----------|----------|----------|----------|----------|----------|----------|----------|
> > > |    |                    | **AWA2** | **CUB**  | **FLO**  |          | **AWA2** |          |          | **CUB**  |          |          | **FLO**  |          |
> > > |    |                    |          |          |          | **u**    | **s**    | **H**    | **u**    | **s**    | **H**    | **u**    | **s**    | **H**    |
> > > | a) | I2D Global         | 69.4     | 37.2     | 37.2     | 59.1     | **79.7** | 67.8     | 28.5     | 59.1     | 38.4     | 28.4     | 88.2     | 43.0     |
> > > | b) | I2D Global + D2I   | 67.1     | 39.5     | 32.0     | 53.9     | 76.5     | 63.2     | 32.0     | **61.4** | 42.1     | 28.3     | 87.0     | 42.7     |
> > > | c) | I2DFormer + D2I    | 68.7     | 42.5     | 37.6     | 58.1     | 76.3     | 66.0     | 32.3     | 52.8     | 40.1     | 34.2     | 86.0     | 48.9     |
> > > | d) | **I2DFormer(Ours)**| **76.4** | **45.4** | **40.0** | **66.8** | 76.8     | **71.5** | **35.3** | 57.6     | **43.8** | **35.8** | **91.9** | **51.5** |
> > >
> > > ***Results:***
> > >
> > > We see from the table that I2DFormer outperforms all baselines across the three datasets. As we compare rows a) and b), the introduction of the D2IAttention module leads to a drop in performance across two datasets due to the information asymmetry in the problem setting as discussed earlier. Row c) improves upon row b) as the model now additionally utilizes our I2DAttention module but its performance is limited by the D2IAttention. Our final model I2DFormer, which only utilizes the I2DAttention module, outperforms all these baselines in row d). Our model is designed with the problem constraints of our ZSL setting and the resulting information asymmetry in mind. While being conceptually simple, it leads to significant performance gains as shown.
> > >
> > >
> > > ### **Q1) How are different input features (e.g. GloVe) used in the Document Transformer**
> > >
> > > We replace the learnable token embedding layer with fixed word embeddings extracted from the models mentioned in Table 3b) followed by a shallow MLP to improve upon this initial representation before input to the learnable Document Transformer. Since we only have limited text while training in the form of documents of seen classes, learning token embeddings from scratch leads to suboptimal performance as unseen documents introduce additional vocabulary. In this data constraint environment, we found the mix of fixed word embeddings plus a shallow MLP to improve the initial representation as a good compromise.

---

> > > > ### Author Response · Authors · 2022-08-02
> > > > **(Response to Reviewer gQFr 4/4)**
> > > >
> > > > ### **Q2) Are any of either the Image or Document Transformer fine-tuned?**
> > > >
> > > > We fix the Image Transformer as pre-trained on ImageNet1K (which respects the GBU split i.e. the pretraining does not include any data of the unseen classes). We learn the Document Transformer from scratch with a fixed word embedding layer as discussed in the earlier point. The choice of this training strategy is motivated by several published works, including [66,B]. Namely, in [66], the authors have studied the impact of freezing, fine tuning, and end to end training of image-text models with noisy text for the zero-shot transfer models like CLIP. The conclusion from these works propose that when dealing with noisy text, it is better to pretrain the image encoder and learn the text encoder from scratch to align the text modality to a common semantic space. Since our training setting has even more noise in the text domain in the form of non-visual facts about the class described in the document, we found this training strategy to perform the best for us.
> > > >
> > > >
> > > >
> > > > ### **Q3) What are the differences between text lengths of different documents?**
> > > >
> > > > Since our text is extracted from web sources to represent descriptions of the class, the text length is as a natural consequence defined by the source of the information. We indeed have to introduce padding and masks to cater for varying sizes of the document for a run time efficient implementation(One could implement it with a for loop for memory efficiency at the cost of run time). However, we found our visual section filtering strategy to reduce the relative differences between the lengths of the different documents. For example, for AWA2, the filtered document is at max 487 tokens long. The mean padding size for this dataset is 34 tokens with a standard deviation of 47. The difference in lengths becomes much more significant if one does not filter the document for visual sections. In such a case ~60% of the document token tensor will be empty/ padded. This further highlights the importance of relatively cheap section filtering for using unstructured text for ZSL.
> > > >
> > > >
> > > >
> > > > ### **Q4) How does the proposed model compare with other ZSL models with respect to computational requirements?**
> > > >
> > > > Our method requires roughly the same computation and training time compared to our closest unsupervised class embedding competitor VGSE[61]. For comparison, our model is trained on a Single A100 GPU and requires only 24 hours to converge to the reported number while utilizing 20GB of VRAM(as also appreciated by Reviewer oTDz). In comparison VGSE takes a similar training time of 25 hours while utilizing 18GB of VRAM.
> > > >
> > > >
> > > >
> > > > Compared to baseline ZSL models like APN and f-VAEGAN-D2, our model requires relatively more training time and GPU memory. However, our model is learning both a ZSL model and a generic unsupervised class embedding that can be utilized by other methods. Once trained, our learned I2DEmb can benefit any of these baseline models as shown in Table 2.
> > > >
> > > >
> > > >
> > > > For model inference, our model’s computational requirements are very similar to the most basic baselines like SJE as we only require a dot product between each image CLS feature and the document CLS features which can be precomputed once for each evaluation run.
> > > >
> > > >
> > > >
> > > > ### **Q5) How are soft attention maps produced?**
> > > >
> > > > The attention matrix learned by the I2DAttention module has the dimension of [image patches x document tokens] (described in manuscript section 3.2). At a patch size of 16x16 for the input image size of 224x224, each document token produces an attention map of 14x14. This attention map is upsampled and overlaid on the image, similar to how attention is visualized in the original ViT paper.
> > > >
> > > >
> > > >
> > > > ### **Q6) differences in classification between s and s_{local} predictions?**
> > > >
> > > > Once the training has been completed, we observe that the two heads have distilled their knowledge to each other with the global head performing slightly better as shown in Supplementary Table 1. We ablated the ensemble of the two heads and found the accuracy of the model to be between the two heads. The errors of the two models tend to overlap with the local head making errors for some cases the global head classified correctly. While being more accurate, the global head is also computationally efficient for inference as it only requires a dot product between the respective CLS features of images and documents.
> > > >
> > > >
> > > >
> > > > ### **Q7) Eq. 4 shouldn’t it be \hat(f)^(pa)?**
> > > >
> > > > Thank you for pointing this out. We have corrected it in the manuscript.
> > > >
> > > >
> > > >
> > > > ### **Minor Issues:**
> > > >
> > > > Thank you for the suggestions, we have corrected these in the updated manuscript.

---

### Official Review · Reviewer_oTDz · 2022-07-17

**Rating:** 7
**Confidence:** 1
**Soundness:** 3 good
**Presentation:** 3 good
**Contribution:** 3 good

**Summary:**

The authors propose a method to learn a joint representation of an image with a very generic description of the object present in the image. This representation makes it possible to associate the discriminating elements of the textual description with visual elements of the image. The objective is to improve the zero-shot learning approaches and to classify images containing objects not seen during the learning phase only from the textual description. The proposed model is composed of two parts: a first part combines an image-based transformer and a text-based transformer through a scoring function that allows to compute the similarity between text embedding and image embedding. A second part allows alignment using visual queries and textual keys in a combined text and image transformer.
Experiments are conducted on standard datasets for zero-shot learning. Wikipedia articles were collected to serve as class descriptions. The collected dataset will be made public after the review process.
Experiments show that the proposed model outperforms state-of-the-art models and allows the classification of images containing objects never seen during the learning process based on their textual description only (unseen classes). Different types of semantic embedding are tested (glove, longformer, mpnet, tfidf). An ablation study and examples of qualitative results are presented.


**Questions:**

I am not familiar enough with the field of zero-shot learning to ask further questions.

**Strengths And Weaknesses:**

I am not familiar with the field of zero-shot learning but it seems to me that the model combining visual queries and textual keys in a transform model is original. The performances of the model seem to exceed the state of the art results and the model obtains interesting performances with simple embedding (Glove), which puts in perspective the contribution of more complex models like longformer for zero-shot learning problems. The bibliography seems to be very complete, again with citations of simple but efficient models (tfidf). The proposed model seems to be able to be trained on a single A100 GPU in one day, which is accessible. The experimental part is very complete with comparison to the state of the art, testing of different embeddings, ablation study and qualitative analysis.

---

> ### Author Response · Authors · 2022-08-02
> **Thank you for the very positive and helpful review!**
>
> We want to thank the reviewer for the very positive feedback. We are glad that the reviewer shares our excitement around the novelty of our idea, the completeness of our experiments, the compute efficiency of our attention module, and our analysis around the performance of modern and classic document embedding models. The reviewer also rated the paper high for soundness, presentation, and contribution.

---

### Author Response · Authors · 2022-08-02
**We want to thank the reviewers for the very helpful reviews!**

We thank the reviewers for their insightful reviews. We have incorporated the very helpful feedback of the reviewers in the updated manuscript and believe it will increase the impact of our work.



**S1)**
The reviewers appreciated the novelty and simplicity of our model. Reviewer oTDz appreciated “model combining visual queries and textual keys in a transform model is original”. Reviewer gQFr “conceptually simple” and “doesn’t require many extra hyperparameters”. Reviewer Sfca “proven effective to capture the relevant image regions based on the collected documents”. Reviewer uvjr “seems natural - using the asymmetric attention(image-query)”.



**S2)**
The reviewers appreciated the performance gains of I2DFormer “exceed the state of the art results - with simple embedding (Glove),” “learned document text features (I2DEmb) quite consistently, both among different datasets and different methods, outperform” “experimental results are considered favorable compared to SOTAs on three benchmark datasets” “The performance over the baselines was improved.”



**S3)**
The reviewers appreciated the interpretability of our learned model “Attention maps provide some extra interpretability “, “The I2D module is proven effective to capture the relevant image regions based on the collected documents.”, “The interpretability study demonstrates that the model learned to align modalities together.”



**S4)**
Reviewers found the experimental evaluation of our work to be “very complete with comparison to the state of the art” with “Rich ablation studies and a good set of experiments”.

In this work, we argue that textual information in the form of class documents serves as natural auxiliary information for zero-shot learning. We propose I2DFormer, a novel transformer-based model that jointly learns image and document embeddings for zero-shot image classification. In addition, our model employs our novel I2DAttention module, which learns fine-grained interaction between image regions and the documents. Our model sets a new SOTA wrt unsupervised class embeddings on three benchmark datasets and additionally offers great interpretability. The class embeddings generated by our model also improve all baselines.

We address individual questions and concerns under each reviewer's comments. Please note that the line numbers referenced in the individual comments refer to the updated rebuttal version of the manuscript and the supplementary.

---

> ### Comment · Reviewer_gQFr · 2022-08-05
> **About the authors response**
>
> I want to thank the authors for their extensive response!
> It addresses my questions and the main points I've made well.
>
> I find the response to other reviewers also generally convincing.

---

### Meta-Review · Area_Chair_2pjD · 2022-08-25

**Recommendation:** Accept
**Confidence:** Less certain

**Metareview:**

The authors propose a method to learn a joint representation of an image with a document of the object present in the image. Experiments show that the proposed model outperforms state-of-the-art models. Although the final reviews between reviewers are not aligned, I think authors solved most of their proposed questions.

**Award:**

No

---

### Decision · Program_Chairs · 2022-09-14

Accept